# Hot-Refresh Model Upgrades with Regression-Alleviating Compatible Training in Image Retrieval

**Binjie Zhang**[1,2‡]   **Yixiao Ge**[2†]   **Yantao Shen**[4‡]   **Yu Li**[5‡]
**Chun Yuan**[1†]   **Xuyuan Xu**[3]   **Yexin Wang**[3]   **Ying Shan**[2]
[1]Tsinghua University   [2]ARC Lab, Tencent PCG   [3]AI Technology Center of Tencent Video
[4]AWS/Amazon AI   [5]International Digital Economy Academy
{zbj19@mails,yuanc@sz}.tsinghua.edu.cn   {yixiaoge,yingsshan}@tencent.com

## Abstract

The task of hot-refresh model upgrades of image retrieval systems plays an essential role in the industry but has never been investigated in academia before. Conventional cold-refresh model upgrades can only deploy new models after the gallery is overall backfilled, taking weeks or even months for massive data. In contrast, hot-refresh model upgrades deploy the new model immediately and then gradually improve the retrieval accuracy by backfilling the gallery on-the-fly. Compatible training has made it possible, however, the problem of model regression[*] with negative flips poses a great challenge to the stable improvement of user experience. We argue that it is mainly due to the fact that new-to-old positive query-gallery pairs may show less similarity than new-to-new negative pairs. To solve the problem, we introduce a Regression-Alleviating Compatible Training (RACT) method to properly constrain the feature compatibility while reducing negative flips. The core is to encourage the new-to-old positive pairs to be more similar than both the new-to-old negative pairs and the new-to-new negative pairs. An efficient uncertainty-based backfilling strategy is further introduced to fasten accuracy improvements. Extensive experiments on large-scale retrieval benchmarks (*e.g.*, Google Landmark) demonstrate that our RACT effectively alleviates the model regression for one more step towards seamless model upgrades[§].

## 1 Introduction

With the rapid development of deep learning, image retrieval algorithms have shown great success in large-scale industrial applications, *e.g.*, face recognition (Sun et al., 2014a;b), object re-identification (Li et al., 2014; Zheng et al., 2015), image localization (Arandjelovic et al., 2016). Model upgrades are essential to improve user experience in those applications. By properly leveraging massive training data and improved network architectures, the retrieval performances could be well boosted.

Generally, once the model updates, the process of "backfilling" or "re-indexing" is called for offline re-encoding the gallery image features with the new model. The retrieval system can only benefit from the new model after the gallery is overall backfilled, which may cost weeks or even months for billions of industry images in practice. The above process is termed *cold-refresh* model upgrades.

To harvest the reward of the new model immediately, Shen et al. (2020) introduces the compatible representation learning to train the new model with backward compatibility constraints, so that queries encoded by the new model can be directly indexed by the old gallery features. Meanwhile, as the new features and old features are interchangeable with each other, the gallery images can be backfilled on-the-fly, and the retrieval performances would be gradually improved to approach the optimal accuracy of the new model, dubbed as *hot-refresh* model upgrades (see Figure 1 (a)).

---

[†]Corresponding authors. [‡]Work done when Binjie, Yantao and Yu are at ARC Lab, Tencent PCG.
[*]Throughout this paper, the term *model regression* indicates the degradation of retrieval performance when the hot-refresh model upgrades proceed.
[§]The code is available at https://github.com/binjiezhang/RACT_ICLR2022.

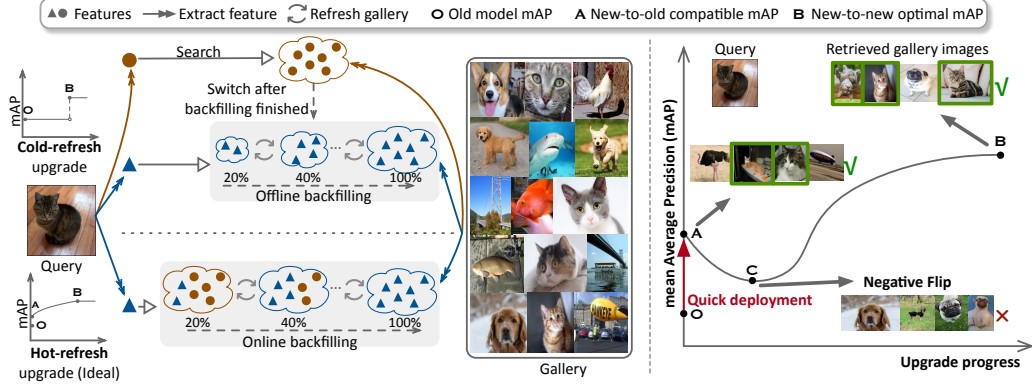

Figure 1: **(a)** The illustration of cold-refresh and hot-refresh model upgrades. **(b)** The problem of model regression with negative flips in hot-refresh model upgrading procedure of image retrieval systems, where queries retrieved correctly by the old model fail during the upgrade process.

Although existing compatible training methods (Meng et al., 2021; Shen et al., 2020) make it possible to upgrade the model in a hot-refresh manner, they still face the challenge of *model regression* (Yan et al., 2020; Xie et al., 2021), which is actually caused by *negative flips*, *i.e.*, queries correctly indexed by the old model are incorrectly recognized by the new model (see Figure 1 (b)). We claimed that, during the procedure of hot-refresh model upgrades, the negative flips occur when the new-to-new similarities between negative query-gallery pairs are larger than the new-to-old similarities between compatible positive query-gallery pairs. To solve the problem, we introduce a **R**egression-**A**lleviating **C**ompatible **T**raining (**RACT**) method to effectively reduce the negative flips for compatible representation learning and further improve the hot-refresh model upgrades with faster and smoother accuracy convergence in image retrieval systems.

Specifically, the conventional backward-compatible regularization is generally designed to pull the new-to-old positive pairs closer and push the new-to-old negative pairs apart from each other. To reduce negative flips during the online gallery backfilling, we improve the backward-compatible regularization by enforcing the new-to-old positive pairs to be more similar than both the new-to-old negative pairs and the new-to-new negative pairs. To further improve the user experience during the model upgrades, we introduce an uncertainty-based hot-refresh backfilling strategy, which re-indexes the gallery images following "the-poor-first" policy. To measure the refresh priority of gallery images in a lightweight manner, we feed the old gallery features into the new classifier (one fully-connected layer in general), yielding the probability distribution on new classes. Even though ground-truth classes may not exist for the old gallery features, we can assume that more discriminative old features are expected to have sharper probability vectors given their compatible latent space. Uncertainty estimation (*e.g.*, Entropy-based) is utilized here to evaluate the sharpness of the probability vectors, and gallery images with larger uncertainty should have higher priority for backfilling. Experimental results on large-scale image retrieval benchmarks (*i.e.*, Google Landmark v2 (Weyand et al., 2020), RParis, and ROxford (Radenović et al., 2018)) demonstrate the effectiveness of our RACT method in alleviating the problem of model regression for hot-refresh model upgrades.

Our contributions can be summarized three-fold. (1) We study the model regression problem in hot-refresh model upgrades of image retrieval systems for the first time. (2) A regression-alleviating backward-compatible regularization is proposed to alleviate the issue of negative flips during compatible training. (3) An uncertainty-based backfilling strategy is introduced to further improve the user experience with "the-poor-first" hot refresh.

## 2    RELATED WORKS

**Compatible Representation Learning.**    The task of compatible representation learning aims at encoding new features that are interoperable with the old features. It breaks the inherent cold-refresh way of model iteration in the industry - the benefit of new models can be immediately reaped in a backfill-free manner. Shen et al. (2020) first formulated the problem of backward-compatible learning and proposed to utilize the old classifier for supervising the compatible feature learning. Budnik & Avrithis (2020) made an effort to reduce the gap between the large model and a small one with-

out the parametric classifier. Wang et al. (2020) introduced a projection head to transform the old features to be compatible with the new ones. Li et al. (2020b) proposed a framework, namely "Feature Lenses", to encourage image representations transformation-invariant. To balance the trade-off between performance and efficiency, Duggal et al. (2021) designed a compatibility-aware neural architecture search scheme to improve the compatibility of models with different sizes. However, since existing compatible algorithms for image retrieval have not investigated the application of hot-refresh model upgrades, the problem of model regression has been overlooked.

**Model Regression.** Model upgrades play a vital role in improving the performance of industrial applications, where the issue of model regression inevitably harms the user experience. Yan et al. (2020) studied the problem of model regression in image classification, where the negative flips indicate the images correctly classified by the old model but misclassified by the new model. A method, namely focal distillation, was introduced to encourage model congruence on positive samples. Srivastava et al. (2020) also investigated the model regression in classification and proposed a new metric to evaluate the regression effects. Xie et al. (2021) probed the model regression problem in natural language processing. The works mentioned above all focused on the classification tasks. Directly employing their methods on the retrieval tasks leads to sub-optimal performances. Moreover, they did not need to consider hot-refresh gallery backfilling in image classification.

**Active Learning.** Our uncertainty-based backfilling strategy is related to active learning. As the real world is teeming with unlabeled data, active learning makes efforts to achieve better performance with fewer annotated samples by choosing valuable samples actively (Settles, 2009). Active learning has been widely adopted in deep learning tasks, *e.g.*, object detection (Holub et al., 2008), text classification (Zhu et al., 2008), and image classification (Cao et al., 2020). Though obtaining the priority of samples is a common issue between active learning and gallery backfilling, the latter focuses more on enjoying benefits in model upgrades with minimal computational overhead.

**Image Retrieval.** The task of image retrieval requires searching images of the same content or object from a large-scale database based on their feature similarities, given images of interest. It is a fundamental task for many visual applications, *e.g.*, face recognition (Guillaumin et al., 2009; Cui et al., 2013), person re-identification (Bak & Carr, 2017; Liu et al., 2017), image localization (Arandjelovic et al., 2016; Ge et al., 2020b). Deep learning-based methods improve the accuracy of image retrieval by casting it as metric learning tasks (Gordo et al., 2016; Chen et al., 2018; Brown et al., 2020), architecture design/search tasks (Zhou et al., 2019), or self-/semi-supervised learning tasks (Ge et al., 2020c;a). Although many efforts have been made to improve retrieval performance, few people pay attention to the bottleneck of model upgrades in industrial applications.

## 3 Hot-Refresh Model Upgrades of Image Retrieval Systems

Given images of interest (referred to as query $\mathcal{Q}$), image retrieval targets correctly recognizing images of the same content or objects from a large-scale database (referred to as gallery $\mathcal{G}$). An image encoder $\phi(\cdot)$ is employed to map the images into feature vectors. Then, the dot product of normalized vectors is used for measuring the similarity between query and gallery images for ranking.

### 3.1 Hot-Refresh Model Upgrades with Compatible Representations

Let $\phi_{\text{old}}$ to be the old image encoder that was deployed in the retrieval system and $\phi_{\text{new}}$ to be a stronger new encoder which is expected to replace the old one. We denote the evaluation metric for image retrieval, *e.g.*, mean Average Precision (mAP), as $\mathcal{M}(\cdot, \cdot)$. The accuracy of the old system is denoted as $\mathcal{M}(\mathcal{Q}_{\text{old}}, \mathcal{G}_{\text{old}})$, where $\mathcal{Q}_{\text{old}}, \mathcal{G}_{\text{old}}$ are the sets of query and gallery features embedded by $\phi_{\text{old}}$, respectively. The new encoder shows an improved accuracy of $\mathcal{M}(\mathcal{Q}_{\text{new}}, \mathcal{G}_{\text{new}})$. In conventional **cold-refresh** model updating pipeline, offline backfilling the gallery images from $\mathcal{G}_{\text{old}}$ to $\mathcal{G}_{\text{new}}$ may take months for the massive data in industry applications, reducing the efficiency of the retrieval system upgrades to a large extent.

Thanks to the compatible representation learning (Shen et al., 2020), query features encoded by the new encoder (*i.e.*, $\mathcal{Q}_{\text{new}}$) can be directly compared to the old gallery features (*i.e.*, $\mathcal{G}_{\text{old}}$). The retrieval system can benefit from the new encoder immediately in a backfill-free manner, significantly accelerating the model iteration at the expense of the optimal performance,

$$\mathcal{M}(\mathcal{Q}_{\text{old}}, \mathcal{G}_{\text{old}}) < \mathcal{M}(\mathcal{Q}_{\text{new}}, \mathcal{G}_{\text{old}}) < \mathcal{M}(\mathcal{Q}_{\text{new}}, \mathcal{G}_{\text{new}}). \tag{1}$$

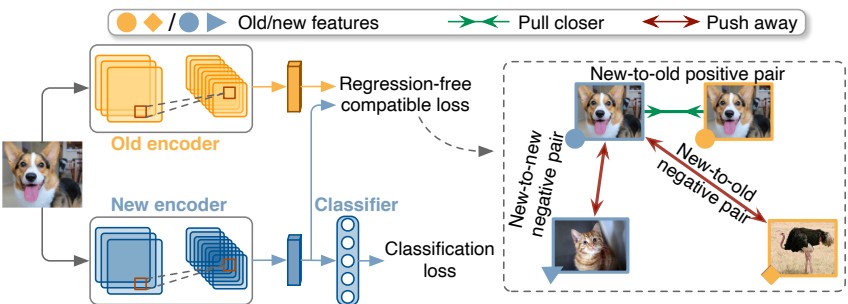

Figure 2: The illustration of our Regression-alleviating Compatible Training (RACT) framework.

Given that the new features and old features are interchangeable due to their compatibility, the gallery images can be backfilled on-the-fly to progressively improve the retrieval accuracy to achieve the optimal performance of the new model,

$$\mathcal{M}(\mathcal{Q}_{\text{new}}, \mathcal{G}_{\text{old}}) \to \cdots \to \mathcal{M}(\mathcal{Q}_{\text{new}}, \mathcal{G}_{\text{new}}^{20\%} \cup \mathcal{G}_{\text{old}}^{80\%}) \to \cdots \to \mathcal{M}(\mathcal{Q}_{\text{new}}, \mathcal{G}_{\text{new}}^{50\%} \cup \mathcal{G}_{\text{old}}^{50\%}) \to$$
$$\cdots \to \mathcal{M}(\mathcal{Q}_{\text{new}}, \mathcal{G}_{\text{new}}^{80\%} \cup \mathcal{G}_{\text{old}}^{20\%}) \to \cdots \to \mathcal{M}(\mathcal{Q}_{\text{new}}, \mathcal{G}_{\text{new}}). \tag{2}$$

The above process is termed as **hot-refresh** model upgrades, where $\mathcal{G}_{\text{new}}^{20\%}$ indicates the 20% of images in the gallery have been re-indexed by the new model $\phi_{\text{new}}$.

## 3.2 Model Regression with Negative Flips

Ideally, the performance of the retrieval system should be smoothly upgraded along with the gallery online backfilling, and the user experience would be continuously improved. However, we observe the phenomenon of **model regression** especially in the early backfilling stages, that is, the accuracy degrades when part of the gallery images are refreshed, *e.g.*,

$$\mathcal{M}(\mathcal{Q}_{\text{new}}, \mathcal{G}_{\text{new}}^{20\%} \cup \mathcal{G}_{\text{old}}^{80\%}) < \mathcal{M}(\mathcal{Q}_{\text{new}}, \mathcal{G}_{\text{old}}). \tag{3}$$

It is actually due to the **negative flips**, where some query images correctly retrieved by the old model are incorrectly indexed by the new model during the upgrade process. There are two main potential factors that cause the negative flips: (1) new-to-old query-gallery positive pairs are less similar than new-to-new query-gallery negative pairs; (2) new-to-new positives are less similar than new-to-old negative pairs. Given that the new model generally encodes more discriminative features than the old model for the overall performance gains, we focus on the former one in this paper, *i.e.*, there exists a query $x_q$ that satisfies

$$\langle \phi_{\text{new}}(x_q), \phi_{\text{old}}(x_g^{\text{pos}}) \rangle < \langle \phi_{\text{new}}(x_q), \phi_{\text{new}}(x_g^{\text{neg}}) \rangle, \tag{4}$$

where $x_g^{\text{pos}}$ is the positive gallery sample, $x_g^{\text{neg}}$ is the negative gallery sample, and $\langle \cdot, \cdot \rangle$ denotes the cosine similarity of two normalized feature vectors. In the case of Eq. (3), $\phi_{\text{new}}(x_q) \in \mathcal{Q}_{\text{new}}$, $\phi_{\text{old}}(x_g^{\text{pos}}) \in \mathcal{G}_{\text{old}}^{80\%}$ and $\phi_{\text{new}}(x_g^{\text{neg}}) \in \mathcal{G}_{\text{new}}^{20\%}$.

## 4 Methodology

We introduce a regression-alleviating compatible training approach (Figure 2) for tackling the challenge of model regression in hot-refresh model upgrades of image retrieval systems, which has essential impacts on the user experience but was ignored by previous compatible training methods. Our key idea is to encourage the new-to-old positive pairs to be more similar than both new-to-old negative pairs and new-to-new negative pairs. In addition, an uncertainty-based strategy is proposed to backfill the gallery in "the-poor-first" manner, in order to fasten accuracy improvements.

### 4.1 Revisit Conventional Compatible Training

Being cast as either the metric learning task (Budnik & Avrithis, 2020) or the transformation learning task (Wang et al., 2020), the core of existing compatible training methods is to encourage the new feature of an image to approach the old features of the intra-class images while keeping away from the old features of different classes. We treat the compatible training as an instance discrimination-like task (*i.e.*, the positive pair is the new and old features of the same instance) for efficient training in large-scale retrieval tasks. To verify that such an instance discrimination-like strategy does not

hurt the retrieval accuracy as well as the feature compatibility, we conduct an experiment by using a pre-sampler for constructing positive pairs of intra-class instances for comparison. As shown in Table 6 of Appendix, similar results can be observed with or without a pre-sampler. The reason for the similar results might be that the most significant objective of compatible training is to train the new model to map images to same features as the old model rather than constraining the distance between positive pairs, according to (Budnik & Avrithis, 2020). Note that the negative pair is built by the new and old features of inter-class instances in the mini-batch, *i.e.*, the intra-class samples would not be treated as the negatives in our instance discrimination-like compatible training. We denote $\mathcal{T}$ as the training set and $\mathcal{B}$ as the mini-batch. Given an anchor image $x \in \mathcal{T}$, we can formulate the backward-compatible regularization as a contrastive loss,

$$\mathcal{L}_{\text{comp}}(x) = -\log \frac{\exp(\langle \phi_{\text{new}}(x), \phi_{\text{old}}(x)\rangle/\tau)}{\exp(\langle \phi_{\text{new}}(x), \phi_{\text{old}}(x)\rangle/\tau) + \sum_{k \in \mathcal{B}\backslash p(x)} \exp(\langle \phi_{\text{new}}(x), \phi_{\text{old}}(k)\rangle/\tau)}, \quad (5)$$

where $\tau$ is a temperature hyper-parameter and $p(x)$ denotes the set of intra-class samples for $x$ (including $x$) in the mini-batch. With the above regularization, $\phi_{\text{new}}$ is trained to embed new features on the same latent space as the old features. However, they ignore the relation between new-old and new-new similarities, facing the risk of negative flips during model upgrading.

## 4.2 Regression-alleviating Compatible Training

**Regression-alleviating Compatibility Regularization.** To constrain the feature compatibility and reduce negative flips at the same time, we improve the conventional backward-compatible loss by adding new-to-new negative pairs as an extra regularization term. Such an improved backward-compatible loss provides regression-alleviating regularization during training, that is formulated as

$$\mathcal{L}_{\text{ra-comp}}(x) = \quad (6)$$

$$-\log \frac{\exp(\langle \phi_{\text{new}}(x), \phi_{\text{old}}(x)\rangle/\tau)}{\exp(\langle \phi_{\text{new}}(x), \phi_{\text{old}}(x)\rangle/\tau) + \sum_{k \in \mathcal{B}\backslash p(x)} \Big( \exp(\langle \phi_{\text{new}}(x), \phi_{\text{old}}(k)\rangle/\tau) + \exp(\langle \phi_{\text{new}}(x), \phi_{\text{new}}(k)\rangle/\tau) \Big)}.$$

In such a way, the new and old compatible positive pairs are trained to achieve more prominent similarities than both new-to-old and new-to-new negative pairs. The issue of negative flips can be alleviated properly when the gallery backfilling proceeds, *i.e.*, the gallery is a mix-up of both old and new features. We do not specifically constrain the new-to-new positive pairs to be closer than new-to-old negative pairs, as the new features are generally trained to be more discriminative than the old ones in order to achieve the model upgrading purpose. Thus, the new-to-new positives are naturally to achieve larger similarities than new-to-old negatives.

**Overall Training Objective.** Besides the compatibility constraints, we also need the ordinary training regularization for image retrieval tasks to maintain the distinctiveness of the new feature representations. Following the strong benchmark in (Weyand et al., 2020), we choose the classification as a pretext task during training. Specifically, we adopt a fully-connected layer as the classifier, denoted as $\omega_{\text{new}} : \mathbf{f} \to \{1, \cdots, C\}$, where $C$ is the number of classes of the new training dataset. The classification loss for the pretext task is formulated as

$$\mathcal{L}_{\text{cls}}(x) = \ell_{\text{ce}}\Big( \omega_{\text{new}} \circ \phi_{\text{new}}(x), y(x)\Big), \quad (7)$$

where $\ell_{\text{ce}}$ denotes the cross-entropy loss, $y(x)$ is the ground-truth label for $x$, and $\omega_{\text{new}} \circ \phi_{\text{new}}(x)$ produces the class logits after a softmax operation. The overall training objective function for our regression-alleviating compatible training framework is,

$$\mathcal{L}(x) = \mathcal{L}_{\text{cls}}(x) + \lambda \mathcal{L}_{\text{ra-comp}}(x), \quad (8)$$

where $\lambda$ is the loss weight. New models trained towards the above objectives enable better hot-refresh upgrades of image retrieval systems by properly alleviating the problem of model regression.

## 4.3 Uncertainty-based Backfilling

In order to further fasten the accuracy convergence during the hot-refresh model upgrades, we propose to refresh the gallery following "the-poor-first" priority, *i.e.*, images with non-discriminative features are expected to be re-encoded first. Given the massive data in industrial applications, we need to quickly measure the distinctiveness of old gallery features in a relatively lightweight way. An uncertainty-based strategy is therefore introduced to measure the priority for backfilling the gallery.

We claim that the confidence of class probabilities represents the feature distinctiveness to some extent. Since the old and new latent spaces are directly comparable after compatible training, we can produce the class probabilities of the old gallery features by feeding them into the trained new classifier $\omega_{\text{new}}$. Although the ground-truth classes of the gallery images are generally non-overlapped with the new training dataset and even unknown, we can assume that more discriminative features should have sharper class probability vectors. Inspired by the "uncertainty sampling" (Settles, 2009), we evaluate the sharpness of the probability vectors using the uncertainty estimation metrics. Samples with larger uncertainty scores will be assigned higher priority to be backfilled. We consider three common ways of measuring uncertainty in this paper.

**Least Confidence Uncertainty.** It considers the gap between $100\%$ and the probability of the predicted class as the uncertainty score for each sample, formulated as

$$\mathcal{U}_{lc}(f_g) = 1 - \text{softmax}[\omega_{\text{new}}(f_g)]^1, \quad f_g \in \mathcal{G}_{\text{old}}, \tag{9}$$

where $\text{softmax}[\cdot]^k$ denotes the probability of rank $k$-th confident class.

**Margin of Confidence Uncertainty.** It measures the uncertainty of class probabilities by the difference between the top-2 confident predictions,

$$\mathcal{U}_{mc}(f_g) = 1 - (\text{softmax}[\omega_{\text{new}}(f_g)]^1 - \text{softmax}[\omega_{\text{new}}(f_g)]^2), \quad f_g \in \mathcal{G}_{\text{old}}, \tag{10}$$

**Entropy-based Uncertainty.** The above metrics only consider the top-1 or top-2 classes, while the Entropy-based uncertainty measurement takes the overall probability distribution into consideration.

$$\mathcal{U}_e(f_g) = -\sum_i^C \text{softmax}[\omega_{\text{new}}(f_g)]^i \cdot \log(\text{softmax}[\omega_{\text{new}}(f_g)]^i), \quad f_g \in \mathcal{G}_{\text{old}}, \tag{11}$$

where $C$ is the number of classes in new training set.

The introduced two technical parts of our work, *i.e.*, regression-alleviating compatible training and uncertainty-based backfilling strategy, together contribute to alleviating the model regression problem in hot-refresh model upgrades. The former **explicitly** regularizes the models to reduce negative flips at training time, while the latter can **implicitly** reduce negative flips by refreshing the poor features (which show a higher risk for negative flips) in priority at test time. They can be considered as an entire solution (from training to testing) for hot-refresh model upgrades.

## 5 EXPERIMENTS

### 5.1 IMPLEMENTATION DETAILS

**Training Data.** Google Landmark v2 (Weyand et al., 2020), GLDv2 in short, is a large-scale public dataset for landmark retrieval. Following the setup in (Jeon, 2020), we use a clean subset of GLDv2, dubbed GLDv2-clean, which consists of 1,580,470 samples with 81,313 classes. In this paper, we investigate three different kinds of data allocations for training the old model and the new model, covering most scenarios of model upgrades in practice. (1) **Expansion**: the new training set is an expansion of the old training set. We randomly sample 30% of images as the old training set, and use the 100% of images as the new training set. (2) **Open-data**: the new training data are excluded from the old training data, but they share the same classes. We randomly sample 30% of each class as the old training set, and use the rest 70% of each class as the new training set. (3) **Open-class**: the new training data and classes are both excluded from the old ones. We randomly sample 30% of classes for the old model training, and images of the rest 70% of classes are utilized for the new model training. The training data details can be found in Table 1.

| Allocation type | Old train-set | | New train-set | |
| --- | --- | --- | --- | --- |
| | # images | # classes | # images | # classes |
| Expansion | 445,419 | 81,313 | 1,580,470 | 81,313 |
| Open-data | 445,419 | 81,313 | 1,135,051 | 81,313 |
| Open-class | 472,604 | 24,393 | 1,107,866 | 56,920 |

Table 1: Three different allocations for the training data, where all the images are sampled from GLDv2-clean. The "expansion" and "open-data" setups share the same old training set.

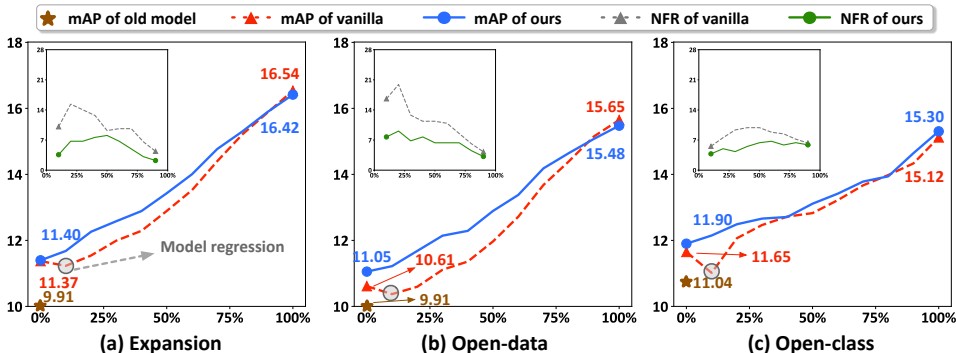

Figure 3: The trend of retrieval performance during the hot-refresh model upgrades (R50-R101) on GLDv2-test, in terms of mAP(↑) and NFR(↓). The vanilla models suffer from the model regression significantly, while ours properly mitigate the model regression to some extent. The results of NFR also indicate the effectiveness of our introduced regression-alleviating compatible regularization.

**Testing Data.** We evaluate the trained models on three test sets, including GLDv2-test, Revisited Oxford (ROxford), and Revisited Paris (RParis). GLDv2-test contains 761,757 gallery images and 750 query images. Following the test protocol in (Radenović et al., 2018), we use the medium testing with 70 queries for both ROxford and RParis. ROxford (Philbin et al., 2007) contains 5,007 gallery images, and RParis (Philbin et al., 2008) consists of 6,357 gallery images.

**Evaluation Metric.** Mean Average Precision (**mAP**) is adopted here for evaluating the retrieval performance. Following the evaluation protocols in (Budnik & Avrithis, 2020; Weyand et al., 2020), we use mAP@100 for GLDv2-test, and mAP@10 for ROxford and RParis. Since mAP measures the overall accuracy with the consideration of both positive flips and negative flips, we define a metric, namely negative flip rate (**NFR**), to measure the degree of model regression in hot-refresh model upgrades. Note that our NFR is specifically defined for image retrieval tasks, different from the one introduced in (Yan et al., 2020). Our NFR is formulated as

$$\text{NFR@}k = \frac{|\hat{\mathcal{Q}}_k^+ \cap \mathcal{Q}_k^-|}{|\hat{\mathcal{Q}}_k^+|}, \tag{12}$$

where $\hat{\mathcal{Q}}_k^+$ denotes the set of queries that correctly recall relative images from the old gallery in top-$k$ candidates. $\mathcal{Q}_k^-$ is the set of queries that fail to be recalled by top-$k$ galleries during upgrading, where the gallery is a mix-up of both old and new features. NFR@1 is used in this paper.

**Architectures.** We study two kinds of architecture combinations, (1) **R50-R101**: ResNet-50 (He et al., 2016) for the old model, and ResNet-101 for the new model; (2) **R50-R50**: ResNet-50 for both old and new models. Due to space limitation, we discuss R50-R101 in the main paper, and the results of R50-R50 can be found in Appendix A.2. Training details can be found in Appendix A.1.

## 5.2 REGRESSION-ALLEVIATING COMPATIBLE TRAINING

We start with the architecture combination of R50-R101, *i.e.*, ResNet-50 for the old model, and ResNet-101 for the new one. To verify that our regression-alleviating compatibility regularization (Eq. (6)) can properly alleviate the issue of model regression, we compare our method to the vanilla compatible training, *i.e.*, $\mathcal{L}_{\text{vanilla}}(x) = \mathcal{L}_{\text{cls}}(x) + \lambda\mathcal{L}_{\text{comp}}(x)$, where $\mathcal{L}_{\text{cls}}$ is defined in Eq. (7), and $\mathcal{L}_{\text{comp}}$ is defined in Eq. (5). Our uncertainty-based backfilling strategy is not used in this section in order to fairly compare with the vanilla settings. The only variable here is the compatibility regularization, *i.e.*, $\mathcal{L}_{\text{comp}}$ for **vanilla** and $\mathcal{L}_{\text{ra-comp}}$ for **ours**. We evaluate the models trained with expansion, open-data and open-class datasets (see Table 1) by testing on three different test sets, *i.e.*, GLDv2-test, ROxford, and RParis, as illustrated in Figure 3, 4, 5, respectively. It can be observed that the models trained with vanilla compatibility regularization (*i.e.*, "vanilla") suffer from the model regression significantly. In contrast, the models trained with our regression-alleviating compatibility regularization (*i.e.*, "ours") mitigate the model regression to some extent by properly reducing the negative flips.

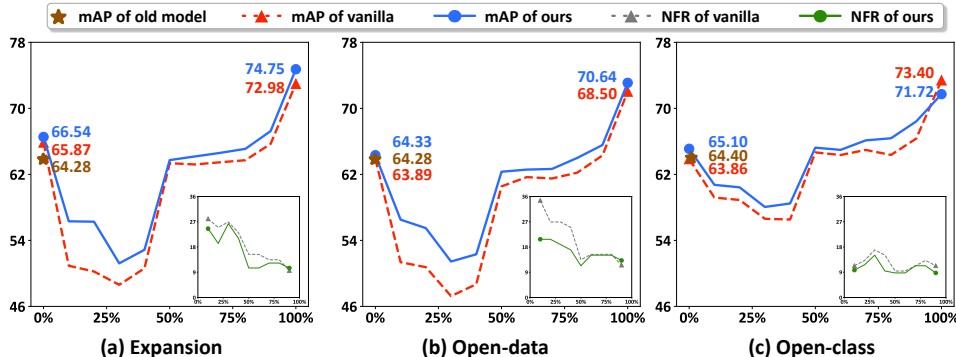

Figure 4: The trend of retrieval performance during the hot-refresh model upgrades (R50-R101) on ROxford, in terms of mAP(↑) and NFR(↓). Evident model regression can be observed for vanilla models, and our method alleviates this issue to a large extent.

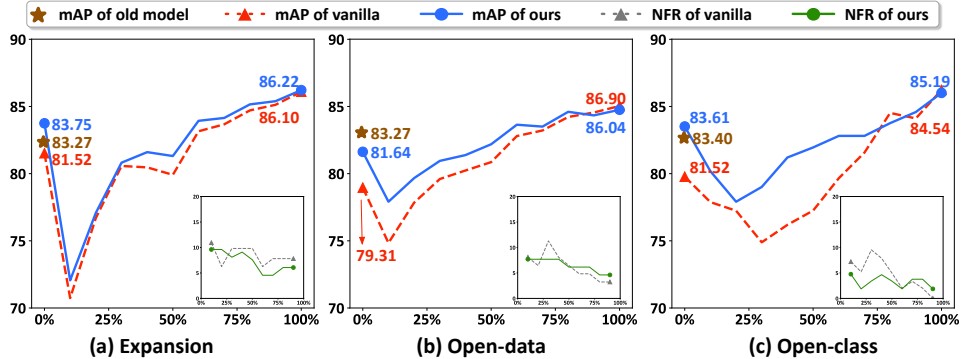

Figure 5: The trend of retrieval performance during the hot-refresh model upgrades (R50-R101) on RParis, in terms of mAP(↑) and NFR(↓). Our method consistently alleviates the issue of model regression on three different training data allocations. The new-to-old compatible mAP at 0% of our model is lower than the old-to-old mAP on the open-data setup, showing the same issue as the vanilla model. It is mainly due to the limited generalization ability of compatible features trained with a sole contrastive loss. Note that in this paper, we aim to study the regression-alleviating regularization rather than advanced compatibility constraints.

## 5.3 UNCERTAINTY-BASED BACKFILLING

The uncertainty-based backfilling strategy is proposed to refresh the gallery following "the-poor-first" priority, in order to fasten the accuracy improvements of the image retrieval during hot-refresh upgrading. The uncertainty-based backfilling is plug-and-play with different models. We discuss different backfilling strategies on ROxford, as shown in Figure 6, and draw the following conclusions: (1) The Least Confidence and Margin of Confidence generally achieve better performance than Entropy-based uncertainty, indicating that the probabilities of top-2 confident classes are more important than the overall classes; (2) The uncertainty-based backfilling strategy can fasten the mAP convergence on top of either vanilla method or our RACT method; (3) Since the uncertainty-based backfilling strategy attempts to refresh the poor features (which show a higher risk for negative flips) in priority, it can indirectly alleviate the model regression problem. How to achieve faster and smoother mAP improvements during backfilling is a new task, and first investigated in this paper.

## 5.4 SEQUENTIAL MODEL UPGRADES

In real-world applications, model upgrades are often conducted sequentially to achieve continuous improvements. To simulate such a scenario, we split the training data into 30%, 60%, 80%, and 100% for training the old model, the upgraded model of the 1st generation, the 2nd generation, and the 3rd generation, respectively. As shown in Figure 7, our method can achieve consistent improvements over the vanilla baseline in terms of hot-refresh model upgrades (solid line). Another kind of model upgrading mechanism (dotted line) introduced by (Shen et al., 2020), denoted as "no-refresh" model upgrades, deploys the compatible new model without refreshing the gallery features.

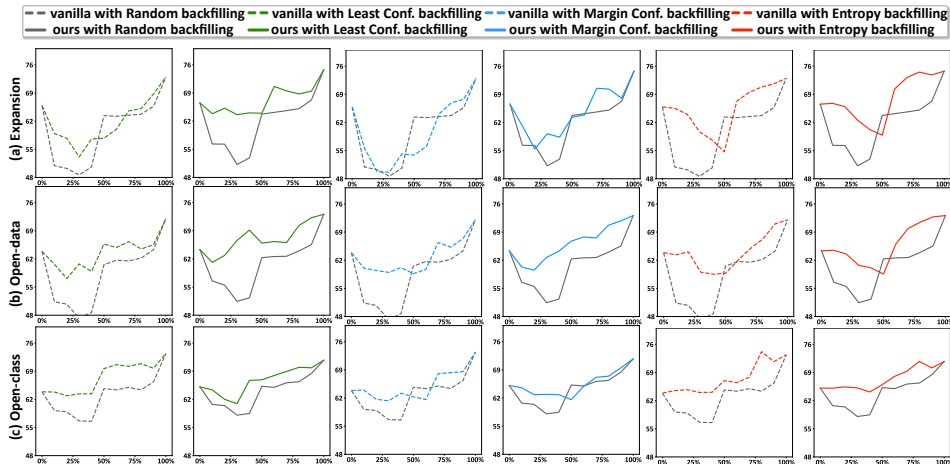

Figure 6: Comparisons between random backfilling and our uncertainty-based backfilling on ROxford. Results of R50-R101 are reported in terms of mAP(↑). Three kinds of uncertainty are studied.

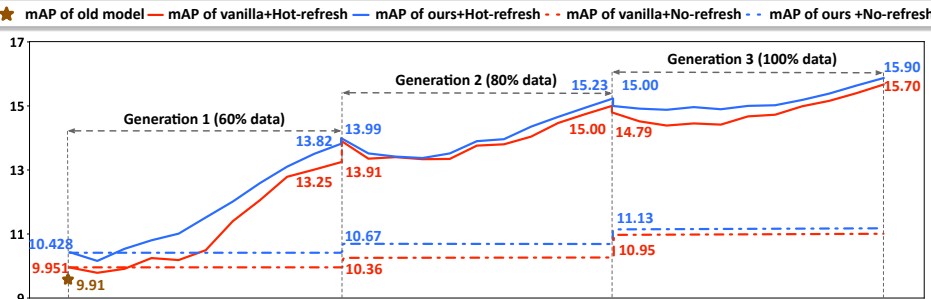

Figure 7: Sequential model upgrades on GLDv2-test (Open-data). Results of R50-R50 are reported in terms of mAP(↑). "Hot-refresh" is the introduced new model upgrading mechanism where the gallery features are refreshed online, and "no-refresh" indicates the backfill-free deployment mechanism introduced by previous compatible learning methods (Shen et al., 2020).

We observe the considerable performances gaps between our hot-refresh model upgrades and the no-refresh model upgrades. It is actually due to the fact that the poor discriminativeness of the old gallery features highly limits the retrieval accuracy of the new model in no-refresh model upgrades.

## 6 CONCLUSION AND DISCUSSION

This paper for the first time studies the issue of model regression during the hot-refresh model upgrades for image retrieval systems. We hope it can arouse more public attention to the bottleneck of efficient model upgrades in industrial applications. Despite that the introduced regression-alleviating compatibility regularization can properly reduce the negative flips to some extent, there is still a long way to go for completely wiping out the problem of model regression. Specifically, several open questions are worth investigating. (1) In this paper, we only consider the negative flips during the gallery backfilling procedure, however, there may also exist negative flips after the backfilling is finished. (2) We investigate regression-alleviating regularization in the form of a plain contrastive loss. It may further help if integrating the regression-alleviating regularization into more advanced backward-compatible losses. (3) We introduce to backfill the gallery in "the-poor-first" priority with uncertainty estimation. It is also a new task for model upgrades. Future works are required for achieving faster and more stable accuracy improvements.

We would also like to discuss the limitations of the introduced method. The raw images of the gallery need to be stored for supporting the hot-refresh model upgrades, which may not be satisfied due to privacy issues. We considered forward-compatible learning (Meng et al., 2021; Wang et al., 2020) as a potential solution for dealing with this limitation, which is expected to upgrade the gallery features with the input of only the old features rather than raw images.

REPRODUCIBILITY STATEMENT

We demonstrate the core of the algorithm in the form of pseudo codes, as shown in Alg. 1 and 2. The training details and hyper-parameters can be found at Appendix A.1. The datasets are all publicly available and the training data splits are carefully illustrated in Table 1. The full code will be released upon acceptance.

---

**Algorithm 1** Pseudocode of Regression-alleviating Compatible Training in a PyTorch-like style.

---

```
# old_model: pretrained and fixed old encoder, no gradient
# new_model: new encoder, new_model.fc is the classifier
# w: loss weight, \lambda in the paper
# tau: temperature, \tau in the paper

for (x, targets) in loader: # load a mini-batch x with N samples
    x_old = aug(x) # a randomly augmented version
    x_new = aug(x) # another randomly augmented version

    with torch.no_grad():
        old_feat = old_model.forward(x_old) # tensor shape: NxD
    new_feat = new_model.forward(x_new)

    # Classification loss, Eqn.(7)
    cls_logits = new_model.fc(new_feat)
    cls_loss = CrossEntropyLoss(cls_logits, targets)

    # select intra-class samples
    masks = targets.expand(N, N).eq(targets.expand(N, N).t())

    # positive logits: Nx1
    l_pos = bmm(new_feat.view(N,1,D), old_feat.view(N,D,1))

    # negative logits: NxN
    # exclude intra-class samples
    l_neg_n2n = mm(new_feat.view(N,D), new_feat.view(D,N)) - masks*1e9
    l_neg_n2o = mm(new_feat.view(N,D), old_feat.view(D,N)) - masks*1e9

    # logits: Nx(1+2N)
    logits = cat([l_pos, l_neg_n2n, l_neg_n2o], dim=1)

    # regression-alleviating compatibility loss, Eqn.(6)
    labels = zeros(N) # positives are the 0-th
    ra_comp_loss = CrossEntropyLoss(logits/tau, labels)

    # SGD update: new model
    loss = cls_loss + w*ra_comp_loss
    loss.backward()
    update(new_model.params)
```

---

bmm: batch matrix multiplication; mm: matrix multiplication; cat: concatenation; t(): matrix transpose.

---

**Algorithm 2** Pseudocode of Calculating Uncertainty-based Backfilling Order in a PyTorch style.

---

```
# N: the number of images in the gallery
# C: the number of classes in the new classifier

def cal_backfilling_order(old_feat, uncertainty_strategy='margin'):
    with torch.no_grad():
        logits = new_model.fc(old_feat) # shape: NxC
    probs = sort(softmax(logits, dim=1), dim=1)

    if uncertainty_strategy == 'least':
        # (1) Least Confidence Uncertainty, Eqn. (9)
        uncertainty_scores = 1 - probs[:,0]
    elif uncertainty_strategy == 'margin':
        # (2) Margin of Confidence Uncertainty, Eqn. (10)
        uncertainty_scores = 1 - (probs[:,0] - probs[:,1])
    elif uncertainty_strategy == 'entropy':
        # (3) Entropy-based Uncertainty, Eqn. (11)
        uncertainty_scores = - sum(probs * log(probs + 1e-9), dim=1)
    else:
        # random backfilling
        return random.shuffle(range(N))

    return argsort(uncertainty_scores) # the-poor-first priority
```

---

sum: sum all elements; log: logarithm; sort: sort the elements in descending order; argsort: the sorted indices in descending order.

ACKNOWLEDGEMENT

This work was supported by NSFC project Grant No. U1833101, SZSTI Grant No. JCYJ20190809172201639 and WDZC20200820200655001, the Joint Research Center of Tencent and Tsinghua.

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

# A APPENDIX

## A.1 TRAINING DETAILS

The networks are pre-trained on the ImageNet dataset (Deng et al., 2009). The images are resized to $224 \times 224$ for both training and testing. During training, random data augmentation is applied to each image before it is fed into the network, including randomly flipping and cropping. We adopt 6 Tesla V100 GPUs for training, and the batch size per GPU is set to 80 for ResNet-50 and 64 for ResNet-101. For all the experiments, Adam optimizer is adopted to optimize the training model with a weight decay of 0.0001. The initial learning rate is set to 0.01 and is decreased to 1/10 of its previous value every 30 epochs in the total 90 epochs. The old model is trained with only a classification loss in Eq. (7), and the new model is trained towards the objective in Eq. (8). The temperature $\tau$ in Eq. (6) is empirically set as 0.05, and the loss weight $\lambda$ in Eq. (8) is set as 1.0.

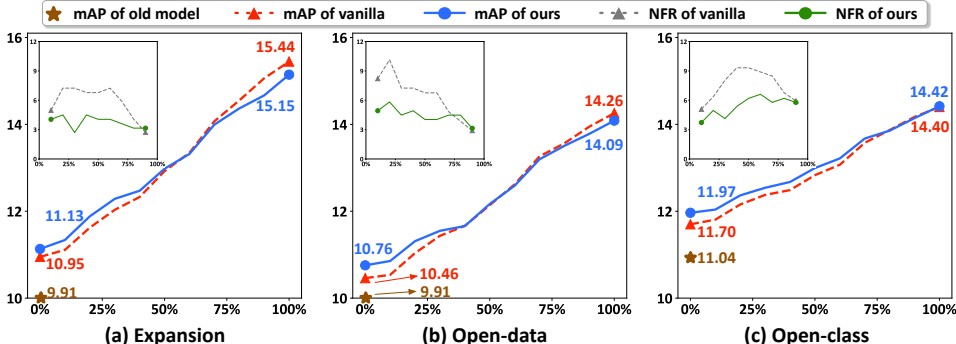

Figure 8: The trend of retrieval performance during the hot-refresh model upgrades of the same architecture, *i.e.*, R50-R50. The results are reported on GLDv2-test in terms of mAP(↑) and NFR(↓). We observe that, despite the overall accuracy (*i.e.*, mAP) has gradually increased without regression, vanilla shows significantly larger NFR than ours. Noted that even if the overall mAP has increased, every new mistake on the old positive queries feels like a step backwards, still leading to a perceived regression in user experience. Reducing NFR can be as important as improving mAP in practice.

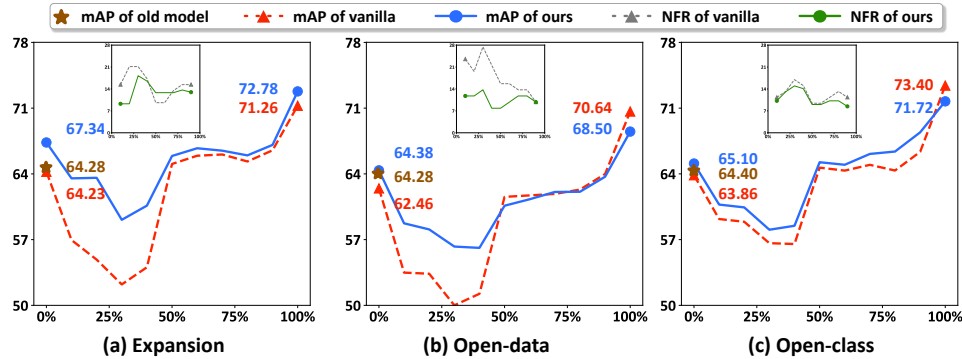

Figure 9: The trend of retrieval performance during the hot-refresh model upgrades (R50-R50) on ROxford, in terms of mAP(↑) and NFR(↓).

## A.2 MODEL UPGRADES OF THE SAME ARCHITECTURE

**Regression-alleviating Compatible Training.** We further evaluate our method when the new model has the same architecture as the old model, *e.g.*, ResNet-50 for both old and new models. As illustrated in Figure 8, the retrieval system with vanilla models is less susceptible to the model regression problem compared to the R50-R101 setup (Figure 3) when testing on GLDv2-test. This is because that old and new models of the same architecture start from the same weights, *i.e.*, ImageNet pre-training. They have natural prediction consistency in the beginning, making the feature

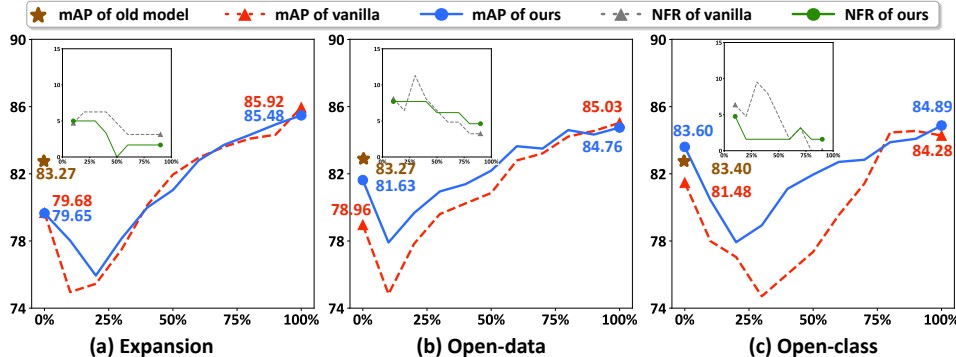

Figure 10: The trend of retrieval performance during the hot-refresh model upgrades (R50-R50) on RParis, in terms of mAP($\uparrow$) and NFR($\downarrow$). Our method consistently alleviates the issue of model regression on three different training data allocations. The new-to-old compatible mAP at 0% of our model is lower than the old-to-old mAP on expansion and open-data setups, showing the same issue as the vanilla model. It is mainly due to the limited generalization ability of compatible features trained with a sole contrastive loss.

compatibility easier. Even so, we argue that vanilla models show evident larger NFR than our model, indicating an underlying model regression. The results of ROxford and RParis are in Figures 9, 10.

**Uncertainty-based Backfilling.** We investigate the uncertainty-based backfilling on the setup of R50-R50, as shown in Figure 11. Backfilling the gallery with uncertainty-based priority can effectively fasten the performance improvements of image retrieval systems, as well as implicitly alleviate the model regression.

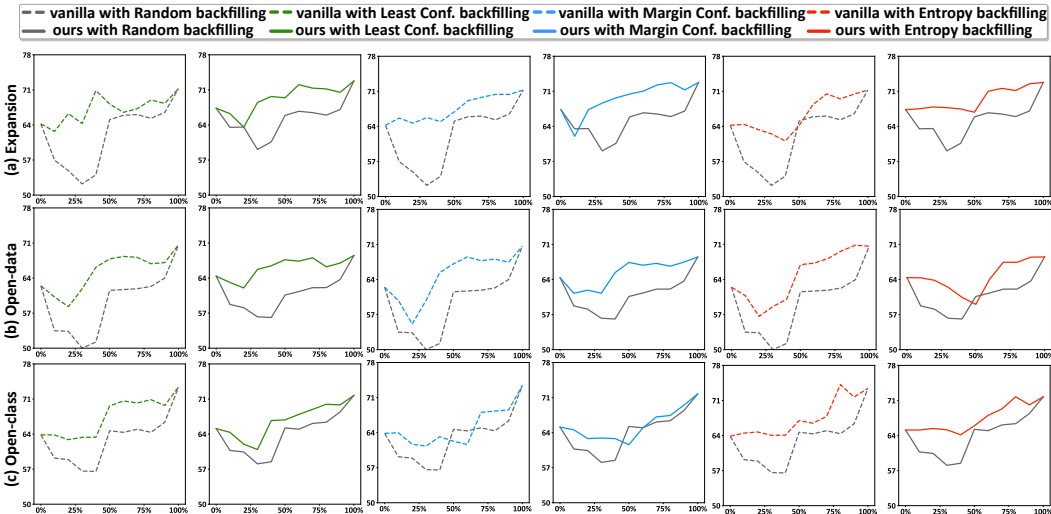

Figure 11: Comparisons between random backfilling and our introduced uncertainty-based backfilling on ROxford, in terms of mAP($\uparrow$). Results of R50-R50 are reported.

## A.3 ADDITIONAL EXPERIMENTS

**Analysis of Uncertainty-based Backfilling.** We further investigate the uncertainty-based backfilling with Margin of Confidence metric on RParis and GLDv2-test, as shown in Figure 12, where the uncertainty-based backfilling generally achieves faster mAP convergence with either the vanilla models or our models. Similar conclusions with the ROxford (discussed in Section 5.3) can be drawn, *i.e.*, the uncertainty-based backfilling strategy can indirectly alleviate the model regression problem since the poor features with a higher risk for negative flips are refreshed in priority.

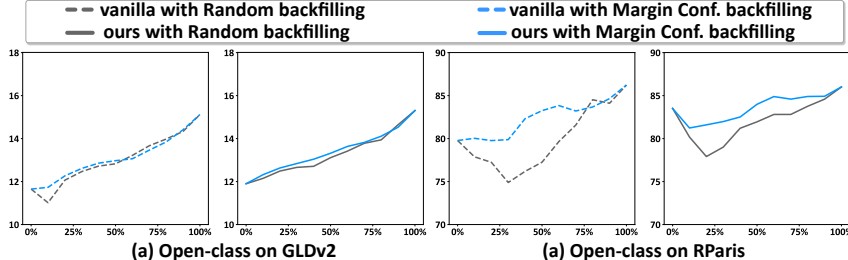

Figure 12: Comparisons between random backfilling and our introduced uncertainty-based backfilling on GLDv2-test and RParis. Results of R50-R101 are reported in terms of mAP(↑).

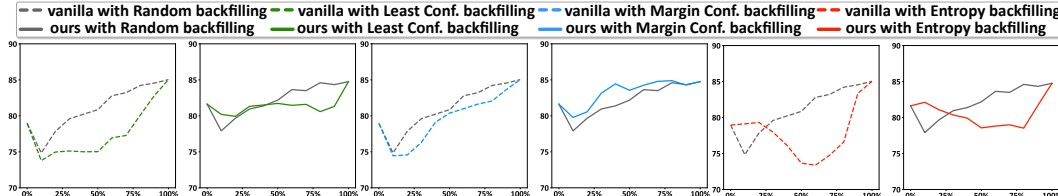

Figure 13: Failure cases of our introduced uncertainty-based backfilling on RParis (Open-data). Results of R50-R101 are reported in terms of mAP(↑).

However, we still face some failure cases with the backfilling strategy, as shown in Figure 13. We argue that the uncertainty-based strategy may not work when the new and old features are not compatible enough, since we use the trained and fixed new classifier to produce class probabilities for the old gallery features under the assumption that they lay on the almost same latent space. To further investigate the phenomenon of Figure 13, we observe that the new-to-old performance (0%) is inferior to the old-to-old performance (brown star) in Figure 5(b) of the main paper, showing the unsatisfactory feature compatibility. More advanced compatible constraints have the potential to solve this issue, but is not the core of this paper.

The introduced uncertainty-based backfilling method is simple but consistently effective in most cases. We acknowledge that the method is not perfect enough to handle all the deployment scenarios, *i.e.*, it fails when the feature compatibility is not good enough (generally when the new-to-old retrieval performance is lower than old-to-old retrieval performance). However, we give the first attempt on a totally new task, showcasing the great potential for further work that incorporates model upgrades with ordered gallery images.

**Compatibility Evaluation.** The evaluation of compatibility can be observed at 0% in all the figures and tables for model upgrades. Specifically, at 0% of model upgrades, the new features of queries need to be directly indexed by the old gallery feature, which is an ordinary benchmark for conventional compatible learning (Shen et al., 2020). To make it clear, we listed the above results in Table 2. We can observe that our method can achieve better compatible performance than the vanilla baseline, indicating that we do not sacrifice the compatibility for reducing negative flips.

| Model | Expansion | | | Open-data | | | Open-class | | |
|---|---|---|---|---|---|---|---|---|---|
| | GLDv2 | ROxford | RParis | GLDv2 | ROxford | RParis | GLDv2 | ROxford | RParis |
| R50 (old) | 9.91 | 64.28 | 83.27 | 9.91 | 64.28 | 83.27 | 11.04 | 64.40 | 83.40 |
| R50 (vanilla new) | 10.95 | 64.23 | **79.68** | 10.46 | 62.46 | 78.96 | 11.70 | 63.86 | 81.48 |
| R50 (ours new) | **11.13** | **67.34** | 79.65 | **10.76** | **64.38** | **81.63** | **11.97** | **65.10** | **83.60** |
| R101 (vanilla new) | 11.37 | 65.87 | 81.52 | 10.61 | 63.89 | 79.31 | 11.65 | 63.86 | 81.52 |
| R101 (ours new) | **11.40** | **66.54** | **83.75** | **11.05** | **64.33** | **81.64** | **11.90** | **65.10** | **83.61** |

Table 2: The evaluation of compatibility in terms of mAP, *i.e.*, 0% results in hot-refresh model upgrades, where the new features of the queries need to be retrieved by the entire old gallery features. It can be considered as an ordinary benchmark for compatible learning.

## A.4 ABLATION STUDIES

**Effects of temperature $\tau$.** The temperature $\tau$ is empirically set as 0.05 in this work. We discuss different values of $\tau$ in Table 3. A similar increasing trend of retrieval performance upgrades can be observed when $\tau$ varies from 0.05 to 0.1, while model regression at 20% is observed when $\tau = 0.01$. The 100% new-to-new performances are sub-optimal when $\tau = 0.1$ and $\tau = 0.08$. In all, 0.05 is a suitable value for the contrastive temperature, and all the experiments in this paper adopt the same temperature value for a fair comparison.

| $\tau$ | old | 0% | 20% | 40% | 60% | 80% | 100% |
|---|---|---|---|---|---|---|---|
| 0.1 | 9.91 | 11.13(+1.22) | 11.44(+0.30) | 11.70(+0.26) | 12.05(+0.35) | 12.57(+0.52) | 13.28(+0.71) |
| 0.08 | 9.91 | 11.24(+1.33) | 11.65(+0.40) | 11.96(+0.31) | 12.29(+0.33) | 12.97(+0.68) | 13.73(+0.76) |
| 0.05 | 9.91 | 11.13(+1.22) | 11.88(+0.75) | 12.47(+0.59) | 13.32(+0.85) | 14.37(+1.05) | 15.14(+0.78) |
| 0.01 | 9.91 | 10.34(+0.43) | 10.07(-0.27) | 10.54(+0.46) | 12.55(+2.01) | 15.12(+2.57) | 15.87(+0.74) |

Table 3: Performance of our method with different values of temperature $\tau$ in terms of mAP($\uparrow$). The results are reported on GLDv2-test with the architectures of R50-R50.

**Effects of loss weight $\lambda$.** The loss weight $\lambda$ is set as 1.0 in this work for simplicity. We study the effects of different $\lambda$ in Table 4. We can observe that the value of $\lambda$ controls the trade-off between new feature discriminativeness and new-to-old feature compatibility, *i.e.*, models with smaller $\lambda$ achieve better performance at 100% while models with larger $\lambda$ achieve better performance at 0%. Despite the start and end points vary from different models, a similar increasing trend of the performances during backfilling can be observed.

| $\lambda$ | old | 0% | 20% | 40% | 60% | 80% | 100% |
|---|---|---|---|---|---|---|---|
| 0.1 | 9.91 | 10.66(+0.75) | 10.66(+0.00) | 11.28(+0.62) | 12.66(+1.38) | 14.34(+1.67) | 15.64(+1.31) |
| 0.5 | 9.91 | 10.94(+1.03) | 11.71(+0.77) | 12.24(+0.53) | 13.19(+0.95) | 14.31(+1.12) | 15.07(+0.76) |
| 1 | 9.91 | 11.13(+1.22) | 11.88(+0.75) | 12.47(+0.59) | 13.32(+0.85) | 14.37(+1.05) | 15.14(+0.78) |
| 2 | 9.91 | 11.10(+1.19) | 11.77(+0.67) | 12.37(+0.60) | 13.16(+0.79) | 13.97(+0.81) | 14.76(+0.79) |
| 5 | 9.91 | 11.24(+1.33) | 11.79(+0.55) | 12.17(+0.38) | 12.60(+0.43) | 13.34(+0.75) | 14.32(+0.97) |

Table 4: Performance of our method with different values of loss weight $\lambda$ in terms of mAP($\uparrow$). The results are reported on GLDv2-test with the architectures of R50-R50.

**Effects of batch size.** We use a batch size of 80 per GPU for the experiments with ResNet-50. As the batch size may affect the number of negative samples for contrastive loss, we conduct ablation studies by modifying the batch size, as illustrated in Table 5. We observe that our method is robust to the number of batch size.

| b.s. | old | 0% | 20% | 40% | 60% | 80% | 100% |
|---|---|---|---|---|---|---|---|
| 40 | 9.91 | 10.98(+1.07) | 11.83(+0.85) | 12.51(+0.68) | 13.53(+1.01) | 14.79(+1.26) | 15.77(+0.99) |
| 60 | 9.91 | 11.05(+1.14) | 11.81(+0.76) | 12.53(+0.71) | 13.43(+0.91) | 14.50(+1.06) | 15.34(+0.85) |
| 80 | 9.91 | 11.13(+1.22) | 11.88(+0.75) | 12.47(+0.59) | 13.32(+0.85) | 14.37(+1.05) | 15.14(+0.78) |

Table 5: Performance of our method with different batch size (b.s.) per GPU in terms of mAP($\uparrow$). The results are reported on GLDv2-test with the architectures of R50-R50. We fail to try larger batch sizes due to the GPU memory limitation.

**Effects of positive sampler.** As mentioned in the method, for more efficient training, we treat the compatible training as an instance discrimination-like task when training with the large-scale retrieval dataset, *i.e.*, GLDv2. Specifically, the positive pair consists of the new and old features of the same instance. To verify that the mentioned instance discrimination-like compatible training does not harm the retrieval performance, we conduct an ablation study by using a positive pre-sampler to randomly sample an intra-class sample for each anchor before each iteration. As shown in Table 6, similar results can be achieved by using either the positive pre-sampler or not. The reason for not seeing an improvement with the positive sampler might be that, in the task of compatible

| Sampler | old | 0% | 20% | 40% | 60% | 80% | 100% |
|---|---|---|---|---|---|---|---|
| w/ | 9.91 | 11.41(+1.50) | 12.11(+0.70) | 12.65(+0.53) | 13.41(+0.76) | 14.33(+0.92) | 15.08(+0.76) |
| w/o | 9.91 | 11.13(+1.22) | 11.88(+0.75) | 12.47(+0.59) | 13.32(+0.85) | 14.37(+1.05) | 15.14(+0.78) |

Table 6: Compare to the results when adopting a positive pre-sampler. The results are reported on GLDv2-test with the architectures of R50-R50, in terms of mAP($\uparrow$). Using the positive pre-sampler achieves slightly better performance at 0%, but lower training efficiency. Similar final performances can be achieved with either the sampler or not, so we discard the positive sampler for more efficient training with large-scale datasets.

training, the most significant objective is to train the new model to map images to same features as the old model, rather than constraining the distance between positive pairs. Such a phenomenon has also been observed by the related work (Budnik & Avrithis, 2020). As shown in Table 3 of (Budnik & Avrithis, 2020), the method of "regression" (forces the old feature and new feature of the same image similar using cosine similarity) wins the asymmetric test (cross-model test), compared to other methods that regularize the positive pairs. Moreover, the training dataset, GLDv2, consists of 1,580,470 samples with 81,313 classes, *i.e.*, 19 samples per class, showing limited intra-class variations, which might also lead to similar results with or without such a positive sampler.

**The trade-off between new-to-old negatives and new-to-new negatives in $\mathcal{L}_{\text{ra-comp}}$.** We investigate the effects of balancing new-to-old negatives and new-to-new negatives in $\mathcal{L}_{\text{ra-comp}}$. Since it does not make sense to directly impose weighted factors on the sample distances (*i.e.*, $\langle\cdot,\cdot\rangle$) in our original loss function (Eq. (6)), we first split our original unified contrastive loss into a combination of two terms, formulated as

$$\mathcal{L}_{\text{ra-comp-split}}(x) = -\Big[ \log \frac{\exp(\langle\phi_{\text{new}}(x),\phi_{\text{old}}(x)\rangle/\tau)}{\exp(\langle\phi_{\text{new}}(x),\phi_{\text{old}}(x)\rangle/\tau) + \sum_{k\in\mathcal{B}\setminus p(x)}\exp(\langle\phi_{\text{new}}(x),\phi_{\text{old}}(k)\rangle/\tau)}$$
$$+ \eta \log \frac{\exp(\langle\phi_{\text{new}}(x),\phi_{\text{old}}(x)\rangle/\tau)}{\exp(\langle\phi_{\text{new}}(x),\phi_{\text{old}}(x)\rangle/\tau) + \sum_{k\in\mathcal{B}\setminus p(x)}\exp(\langle\phi_{\text{new}}(x),\phi_{\text{new}}(k)\rangle/\tau)}\Big], \quad (13)$$

where $\eta$ is a weighting factor. When setting $\eta = 1$, similar results can be observed as the original setup. When changing $\eta$ from 0.2 to 5.0, we find that a larger $\eta$ may lead to better alleviation of model regression but lower new-to-old compatible performance (0%). We find $\eta = 1.0$ a better trade-off between feature compatibility and alleviating model regression, showing that our unified version (Eq. (6)) of contrastive loss is reasonable. Detailed results are illustrated in Figure 14.

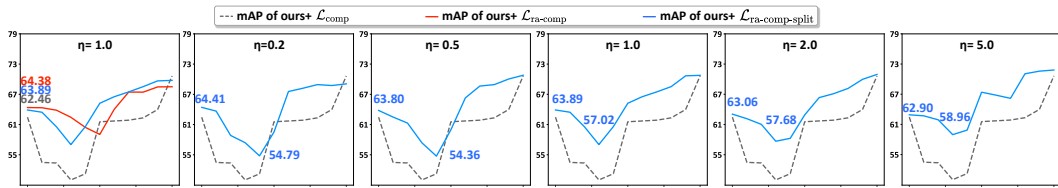

Figure 14: The analysis of trade-off between new-to-old negative pairs and new-to-new negative pairs in our regression-alleviating compatible regularization. We report retrieval performance during the hot-refresh model upgrades (R50-R50) on ROxford in terms of mAP($\uparrow$).

**Different temperatures for new-to-old negatives and new-to-new negatives in $\mathcal{L}_{\text{ra-comp}}$.** We empirically set the temperature to 0.05. To further analyze the effects of different temperatures in two types of negative pairs, we conduct experiments as shown in Table 7. Fewer effects can be observed when changing the temperature, so we use 0.05 for both for brevity.

**An alternative form of compatible regularizations - triplet loss.** In the main paper, we introduce to regularize the compatibility of the learned representations in the form of a contrastive loss. An alternative option for formulating the regularization is in the form of a triplet loss, where several triplets combining positives and negatives from old and new models are considered at the same time. Specifically, the vanilla version of backward-compatible regularization is formulated as,

$$\mathcal{L}_{\text{comp}}(x) = \frac{1}{|\mathcal{B}\setminus p(x)|} \sum_{k\in\mathcal{B}\setminus p(x)} \Big( \max(\langle\phi_{\text{new}}(x),\phi_{\text{old}}(x)\rangle - \langle\phi_{\text{new}}(x),\phi_{\text{old}}(k)\rangle + m, 0)\Big), \quad (14)$$

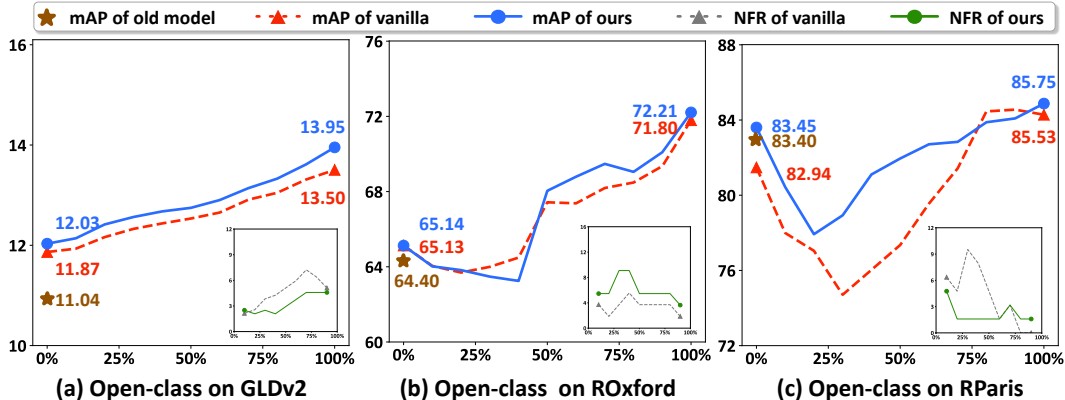

Figure 15: Experiments of compatibility constraints in the form of a triplet loss. We illustrate the trend of retrieval performance during the hot-refresh model upgrades (R50-R50) on GLDv2-test, ROxford and RParis datasets, in terms of mAP(↑) and NFR(↓).

| $\tau_1$ | $\tau_2$ | old | 0% | 20% | 40% | 60% | 80% | 100% |
|------|------|------|------|------|------|------|------|------|
| 0.05 | 0.05 | 9.91 | 10.76(+0.85) | 11.31(+0.56) | 11.66(+0.35) | 12.59(+0.93) | 13.51(+0.92) | 14.09(+0.58) |
| 0.05 | 0.1 | 9.91 | 10.64(+0.73) | 11.14(+0.50) | 11.51(+0.36) | 12.53(+1.03) | 13.43(+0.90) | 14.12(+0.69) |
| 0.1 | 0.05 | 9.91 | 10.56(+0.65) | 11.15(+0.59) | 11.52(+0.36) | 12.43(+0.91) | 13.47(+1.05) | 13.89(+0.42) |

Table 7: Performance of our method with different temperature in Eq. (6) in terms of mAP(↑). The results are reported on GLDv2-test (Open-data) with the architectures of R50-R50. $\tau_1$ and $\tau_2$ denote the temperature for new-to-old negative pairs and new-to-new negative pairs, respectively.

| Sampler | old | 0% | 20% | 40% | 60% | 80% | 100% |
|------|------|------|------|------|------|------|------|
| w/ | 9.91 | 10.76(+0.85) | 11.31(+0.56) | 11.66(+0.35) | 12.59(+0.93) | 13.51(+0.92) | 14.09(+0.58) |
| w/o | 9.91 | 10.82(+0.91) | 11.30(+0.48) | 11.77(+0.47) | 12.59(+0.82) | 13.51(+0.92) | 14.12(+0.61) |

Table 8: Compare to the results when adopting a positive pre-sampler for experiments of triplet loss. The results are reported on GLDv2-test with the architectures of R50-R50, in terms of mAP(↑).

where the margin $m$ is empirically set to $0.8$ and $p(x)$ denotes the set of intra-class samples for $x$ (including $x$) in the mini-batch. The regression-alleviating compatibility regularization therefore becomes

$$\mathcal{L}_{\text{ra-comp}}(x) = \frac{1}{|\mathcal{B} \setminus p(x)|} \sum_{k \in \mathcal{B} \setminus p(x)} \Big( \max(\langle \phi_{\text{new}}(x), \phi_{\text{old}}(x) \rangle - \langle \phi_{\text{new}}(x), \phi_{\text{old}}(k) \rangle + m, 0) +$$

$$\max(\langle \phi_{\text{new}}(x), \phi_{\text{old}}(x) \rangle - \langle \phi_{\text{new}}(x), \phi_{\text{new}}(k) \rangle + m, 0) \Big). \qquad (15)$$

The comparisons between vanilla method and our regression-alleviating method in terms of open-class data split on three different datasets can be found in Figure 15. We observe that our method can properly reduce negative flips on GLDv2 and RParis datasets, but fail in ROxford. Both vanilla and ours show inferior performance than the contrastive counterparts, since the unified form of a contrastive loss can establish a global structure among new-to-old and new-to-new pairs while dividing them into several triplets cannot.

We also investigate the effects of a positive pre-sampler when conducting experiments with triplet loss. As demonstrated in Table 8, similar results can be found with or without such a sampler, drawing the same conclusion with the experiments of contrastive loss.

## A.5  RELEVANT TOPICS

We discuss three relevant topics that also need an old model for training the new model, including compatible learning (Shen et al., 2020; Meng et al., 2021; Budnik & Avrithis, 2020), knowledge distillation (Hinton et al., 2015; Li et al., 2020a; Yan et al., 2020), and continual learning (Li & Hoiem, 2017; Wang et al., 2021).

**Compare to Compatible Learning.** Conventional compatible learning methods (Shen et al., 2020; Meng et al., 2021; Budnik & Avrithis, 2020) regularize the feature compatibility by enforcing the new-to-old positive pairs to be more similar than new-to-old negative pairs. They did not consider the task of hot-refresh model upgrades thus overlooked the problem of negative flips. We take the state-of-the-art BCT (Shen et al., 2020) as an example. The comparisons can be found in Figures 16 and 17. As it only focuses on the backfill-free model upgrades, it cannot alleviate the negative flips, suffering from the problem of model regression. Note that BCT adopts classification loss as compatible constraints (via feeding the new features into the old classifier), where it is hard to explicitly integrate the regularization of new-to-new negative pairs into the form of BCT (classification) loss. So we use our regression-alleviating compatible loss on top of BCT loss, and achieve state-of-the-art compatible performance at the same time alleviating model regression.

**Compare to Knowledge Distillation.** Although we both regularize the prediction consistency between two different models, we have entirely different purposes. Specifically, distillation aims at training a new model that inherits the knowledge of the teacher (old) model in order to improve the overall accuracy of the new model, and the new model will be deployed individually during inference. In contrast, our task requires cross-model retrieval via learning compatible features at the same time reducing negative flips. Well noted that learning new features that are interoperable with old features is not equal to inheriting the knowledge of old features. Moreover, we have different experimental setups. In knowledge distillation, the old model is generally a stronger network with more powerful representation discrimination. In our model upgrading task, the old model is generally a poorer network with inferior representation discrimination. To verify that existing distillation-based algorithms are inapplicable for hot-refresh model upgrades of retrieval tasks, we compared to related work, namely Focal Distillation (FD) (Yan et al., 2020), which was specially designed for alleviating the model regression of classification tasks via knowledge distillation. As shown in Figures 16 and 17, directly employing Focal Distillation in our task achieves unsatisfactory retrieval performances and is hard to achieve the goal of model upgrades. It is actually due to the fact that it inherits the knowledge of the inferior old model too much, which limits its capability of learning discriminative new features.

**Compare to Continual Learning.** Our work is also related to continual learning (Li & Hoiem, 2017; Wang et al., 2021), which aims at training a new model that inherits the capability of the old task. We mainly differ in two ways. (1) Application: Hot-refresh model upgrade requires feature compatibility for cross-modal retrieval, *i.e.*, the query is embedded by the new model while the gallery is a mix of both new and old features. Continual learning requires the new model to remember the capability of the old model task during training, however, the new model is individually deployed without the "interaction" with the old model. Moreover, since continual learning does not need to conduct cross-model deployment, there's no consideration of negative flips. (2) Purpose: Hot-refresh model upgrade requires the new model to outperform the old model and the new features to be interoperable with old features. Continual learning only requires the new model to inherit the old task knowledge, does not expect the new model to outperform the old model on the old task.

To verify that existing continual learning algorithms are inapplicable for hot-refresh model upgrades of retrieval tasks, we compared to related work, namely Learning without Forget (LwF) (Li & Hoiem, 2017). As shown in Figure 18, significant model regression and unsatisfactory retrieval performances are observed. Existing continual learning algorithms cannot tackle the model regression problem of hot-refresh model upgrades.

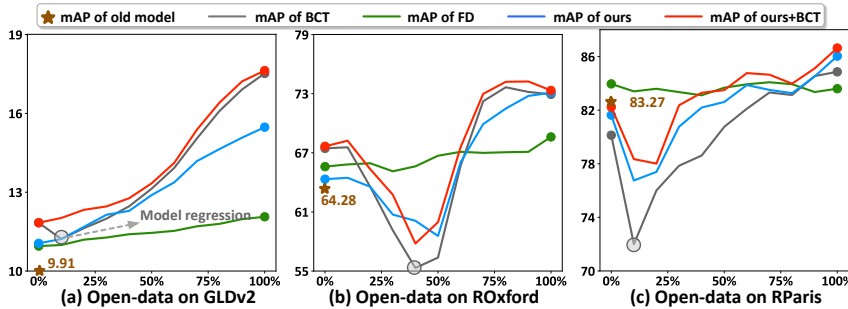

Figure 16: Comparison with two related methods, BCT (Shen et al., 2020) and Focal Distillation (FD) (Yan et al., 2020). The former one only regularizes the feature compatibility while ignoring model regression, and the latter one is specially designed for model regression of classification tasks. We achieve the optimal performance by using our regression-alleviating compatible loss on top of BCT. Results of R50-R101 are reported in terms of mAP(↑).

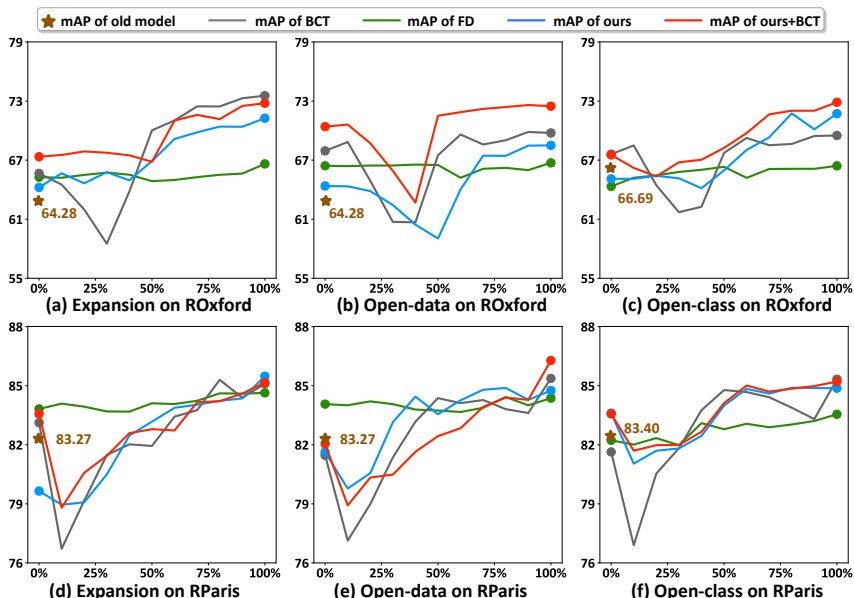

Figure 17: Comparison with two related methods, BCT (Shen et al., 2020) and Focal Distillation (FD) (Yan et al., 2020). Results of R50-R50 are reported in terms of mAP(↑).

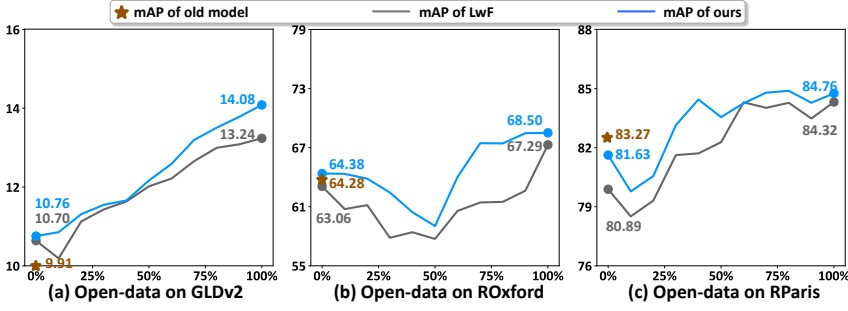

Figure 18: Comparison with continual learning method LwF (Li & Hoiem, 2017) on GLDv2-test, ROxford and RParis. Results of R50-R50 are reported in terms of mAP(↑).

