# OpenReview forum: "Hot-Refresh Model Upgrades with Regression-Free Compatible Training in Image Retrieval"
_ICLR.cc/2022/Conference — ICLR 2022 Poster_

### Official Review · Reviewer_QLFh · 2021-11-01

**Correctness:** 3
**Technical Novelty And Significance:** 3
**Empirical Novelty And Significance:** 3
**Recommendation:** 6
**Confidence:** 4

**Main Review:**

Strengths:
S1) First to study the impact of such “hot-refresh” techniques to the image retrieval domain, relying on standard datasets in this area.
S2) The observation about the reason behind negative flips is novel and interesting – it had not been discussed in previous work. The proposed training method is well-motivated from this observation.
S3) The proposed training method achieves improved performance in the hot-refresh application compared to previous work, reducing the penalty incurred by negative flips.

Weaknesses:
W1) The uncertainty-based backfilling approach does not consistently show performance improvements. I am not convinced by this part. In some cases, random backfilling even performs better. The proposed uncertainty-based backfilling method overall feels simplistic.
W2) Writing could be improved. For example:
Page 4, “We figure out the reason for the negative flips (...)” – the writing here is a little informal. This part could also be rewritten for more clarity / better explanation. It took me some time to parse why this was the only reason for negative flips – this should be explained better.
Page 8, in section 5.2. The paragraph mentions the figures, but does not discuss the results shown in them at all.
W3) The method still shows room for improvement. For example, the lack of pre-sampler to include positive image pairs in the same batch probably hinders performance (the authors justify the lack of that due to it being “hard”, although this is common in deep metric learning, eg for the triplet loss).

Minor details:
M1) Page 5, below Eq 7: “denotes the cross-entropy” — I think the authors here mean softmax + cross-entropy, right?
M2) The Oxford and Paris dataset versions used are more commonly referred to as Revisited Oxford (ROxford) and Revisited Paris (RParis). The renaming would make the dataset usage more clear for the readers.


**Summary Of The Paper:**

The paper proposes a new technique for updating image retrieval models while not requiring full backfill of the index in order to allow deployment. This has the potential to make model upgrades more practical for real-world large-scale applications. The paper is the first to study the impact of such “hot-refresh” techniques to the image retrieval domain. The authors make observations about the reason behind negative flips, which motivate their newly-proposed loss function / training technique. The paper also proposes an uncertainty-based backfilling method to decide the ordering in which images in the index are backfilled. Experiments are conducted on standard image retrieval datasets.

**Summary Of The Review:**

This is an interesting paper, targeting an important application, proposing a novel observation and method to improve hot-refresh image retrieval. However, some aspects of the paper are not well-developed (see weaknesses listed above). Due to the latter, my rating is “marginally above the acceptance threshold”.

---

> ### Author Response · Authors · 2021-11-19
> **Responses to Reviewer QLFh  (1/2)**
>
> Thank you for the valuable and positive comments! We really appreciate the comments for improving the clarity of statements and experimental verifications. The manuscript is revised accordingly, and the main concerns are listed below.
>
> **Q1:** The uncertainty-based backfilling approach does not consistently show performance improvements. I am not convinced by this part. In some cases, random backfilling even performs better. The proposed uncertainty-based backfilling method overall feels simplistic.
>
>
>
> **A1:**
>
> For clearer comparisons, we took different uncertainty-based backfilling strategies apart, as illustrated in Figure 6/11/12 (Page 9,14) in the revised manuscript. From Figure 6/11/12, we can observed that:
>
> (i) Least Confidence and Margin of Confidence show more stable benefits compared to Entropy-based method, indicating that the probabilities of top-2 confident classes are more important than the overall classes;
>
> (ii) The uncertainty-based backfilling strategy can fasten the mAP improvements on top of either vanilla method or our method;
>
> (iii) Since the uncertainty-based backfilling strategy attempts to refresh the poor features (which show a higher risk for negative flips) in priority, **it can indirectly alleviate the model regression problem**.
>
> We acknowledge that there exist failure cases, as illustrated in Figure 13 of the Appendix (Page 15). The uncertainty-based strategy may not work when the new and old features are not compatible enough, since we use the trained and fixed new classifier to produce class probabilities for the old gallery features under the assumption that they lay on the almost same latent space. To further investigate the phenomenon of Figure 13, we observe that the new-to-old performance (0%) is inferior to the old-to-old performance (brown star) in Figure 5(b) of the main paper, showing the unsatisfactory feature compatibility. We carefully discuss such a limitation for our introduced uncertainty-based backfilling strategy in Appendix A.3 (Page 14-15).
>
> To conclude, the introduced uncertainty-based backfilling method is simple yet consistently effective in **most** cases. We acknowledge that the method is not perfect enough to handle all the deployment scenarios, i.e., it fails when the feature compatibility is not good enough (generally when the new-to-old retrieval performance is lower than old-to-old retrieval performance). However, we give the first attempt on a totally new task, showcasing the great potential for further work that incorporates model upgrades with ordered gallery images.

---

> > ### Author Response · Authors · 2021-11-19
> > **Responses to Reviewer QLFh  (2/2)**
> >
> > **Q2:** Writing could be improved.
> >
> > **(a)** For example: Page 4, “We figure out the reason for the negative flips (...)” – the writing here is a little informal. This part could also be rewritten for more clarity / better explanation. It took me some time to parse why this was the only reason for negative flips – this should be explained better.
> >
> > **(b)** Page 8, in section 5.2. The paragraph mentions the figures, but does not discuss the results shown in them at all.
> >
> > **(c)** Page 5, below Eq 7: “denotes the cross-entropy” — I think the authors here mean softmax + cross-entropy, right?
> >
> > **(d)** The Oxford and Paris dataset versions used are more commonly referred to as Revisited Oxford (ROxford) and Revisited Paris (RParis). The renaming would make the dataset usage more clear for the readers.
> >
> >
> >
> > **A2:** Thanks for the valuable advice. We have revised our manuscript following your suggestions (marked in blue).
> >
> > **(a)** We discussed more on the reasons for the negative flips in Page 4: There are two main potential factors that cause the negative flips: (1) new-to-old query-gallery positive pairs are less similar than new-to-new query-gallery negative pairs; (2) new-to-new positives are less similar than new-to-old negative pair. Given that the new model generally encodes more discriminative features than the old model for the overall performance gains, we focus on the former one in this paper.
> >
> > **(b)** We added the discussion of the figures in Section 5.2.
> >
> > **(c)** Yes. We refined it in the manuscript.
> >
> > **(d)** Thanks for the valuable advice.  We have revised the manuscript accordingly.
> >
> >
> >
> >
> >
> > **Q3:** The method still shows room for improvement. For example, the lack of pre-sampler to include positive image pairs in the same batch probably hinders performance (the authors justify the lack of that due to it being “hard”, although this is common in deep metric learning, eg for the triplet loss).
> >
> >
> >
> > **A3:**  Sorry for the confusion. Actually, we conducted the ablation study with a pre-sampler in the first version (Table 5 of Appendix), and similar results were observed with or without such a sampler. We have referenced this experiment in the main paper (Section 4.1 in blue), and added more detailed explanations in the Appendix A.4 (Table 6).
> >
> > The reason for not seeing an improvement with the positive sampler might be that, in the task of compatible training, the most significant objective is to train the new model to map images to similar features as the old model, rather than constraining the distance between positive pairs. Such a phenomenon has also been observed by the related work [1]. As shown in Table 3 of [1], the method of regression (forces the old feature and new feature of the same image similar) wins the asymmetric test (cross-model test), compared to other methods that regularize the positive pairs. Moreover, the training dataset, GLDv2, consists of 1,580,470 samples with 81,313 classes, i.e., ~19 samples per class, showing limited intra-class variations, which might also lead to similar results with or without such a positive sampler.
> >
> >
> >
> > Ref:
> >
> > [1] Mateusz Budnik and Yannis Avrithis. Asymmetric metric learning for knowledge transfer. CVPR 2021.

---

> > > ### Comment · Reviewer_QLFh · 2021-11-25
> > > **Response to author rebuttal's 2/2**
> > >
> > > Q2. Thanks for the fixes.
> > >
> > > Q3. Thanks for the careful explanations. I appreciate that this discussion was added to the paper.

---

> > ### Comment · Reviewer_QLFh · 2021-11-25
> > **Response to author rebuttal's 1/2**
> >
> > I appreciate the new results and discussion, and the authors making it clear that this method does fail in some cases. I agree that the method can show some benefits, although I still think the method is rather simplistic -- a heuristic that works most of the time, though not extremely consistently.

---

### Official Review · Reviewer_Fu6z · 2021-11-02

**Correctness:** 3
**Technical Novelty And Significance:** 2
**Empirical Novelty And Significance:** 3
**Recommendation:** 6
**Confidence:** 3

**Main Review:**

## Strengths

First of all, this paper is mostly well-written and well-motivated, even though there are some parts where the readability could be further improved, especially in terms of definition and references on key terms/ideas such as “backfilling” “cold/hot-refresh”, “negative flip” etc. However, this does not impact the overall readability of the paper too much in my opinion.

For the results, I think they are clearly presented and link to the contributions well, as demonstrated in Figures 3-7 which shows that the new losses and uncertainty-driven backfilling indeed helping the hot-refresh model update process significantly. I particularly appreciate the small figures in Figures 3-5, which shows the progress in NFT v.s. vanilla, as it ties back to the observation of negative flips causing the model regression problem as a main motivation for this work.

## Weaknesses
Firstly, in terms of novelty, I think this work is ok as neither of these techniques have been used for model updates in image retrieval to the best of my knowledge. However, for uncertainty based backfilling, the authors have also mentioned that the techniques are based on Settles, 2009, which is on active learning.

The other main contribution is in the Regression-Free Compatibility Regularization (RFCT) in Eq. 6, as mentioned above, alleviates negative flips by diluting the significance of both new-to-old and new-to-new negative pairs in the contrastive loss. This is where I find the contribution could have much more potential, as this simple addition to the loss could have a lot more variety resulting in much more in-depth analysis of the impact of this term - e.g. changing the ratio between the two terms in the loss? Or perhaps different temperatures in each of them? Or even some dynamic tuning of these hyperparameters according to the uncertainty in the backfilling process, which ties the two novelties together. I think there is so much more that can be done to strengthen this novelty, but unfortunately is not presented in this work.

Another comment I have is the phrase “Regression-free”. If I understand correctly from the authors’ definition, the results in Figure 7 shows that this work is only regression free on the GLDv2 dataset (which the vanilla baseline is already close to regression free), and there is still significant model-regression observed in Oxford5k and Paris6k. I think in this case, claiming the main contribution of this work as “regression-free” is slightly misleading and I would suggest toning down the name to something less strong, e.g. “regression-alleviating”.

Last but not least, not really a weakness here but just a general question, as the authors also took inspiration from active learning, I am wondering how does “hot-fresh” model upgrades compare against other techniques such as continual learning? For example, *Continual Learning for Image-Based Camera Localization* (Wang et. al 2021) also tackles the problem of learning gradually on unseen data, but on the problem of visual localization (which is still quite related to image retrieval). As I am not an expert on active learning / model upgrades, it would be great if the authors could clarify this.


**Summary Of The Paper:**

This paper tackles the problem of model upgrades in large-scale training of image retrieval models. It proposes a new framework of “hot-refresh” updates with two main techniques:
1) by alleviating “negative flip” by a regression-free regularization term in the loss and
2) by prioritizing gallery images with the highest uncertainties during the backfilling process.

The results suggest improvement across three major retrieval benchmarks compared against baseline model-upgrade methods.

**Summary Of The Review:**

Overall, this paper is a good read and does provide useful insights to the proposed problem and potential remedies.
However, I think this works is just so slightly short of passing my threshold for acceptance due to the lack of exploring more on its main contribution, which is the RFCT loss, and potentially giving more valuable information to the community on the synergies between old and new features in the model upgrade process.
Moreover, the slight overclaim in the "Regression-Free" naming requires some extra attention.
Therefore, I am recommending "5: marginally below the acceptance threshold" now, but this decision is marginal and I am willing to consider boosting the score if the authors address my concerns well in the rebuttal.

---

> ### Author Response · Authors · 2021-11-19
> **Responses to Reviewer Fu6z (1/3)**
>
> Thank you for the constructive comments for improving our clarity of statements and experimental verifications. The manuscript is revised accordingly, and the responses to your main concerns are listed below.
>
>
>
> **Q1:** Firstly, in terms of novelty, I think this work is ok as neither of these techniques have been used for model updates in image retrieval to the best of my knowledge. However, for uncertainty based backfilling, the authors have also mentioned that the techniques are based on Settles, 2009, which is on active learning.
>
>
>
> **A1:** Thank you first for acknowledging the novelty of the introduced model upgrade problem. The sub-topic of quickly obtaining the backfilling priority is also **introduced for the first time**, and we propose a simple yet effective uncertainty-based method as the first attempt.
>
> Although the metrics for measuring the uncertainty of class probabilities are based on existing techniques, **the idea of receiving the "pseudo" class probabilities of massive old gallery features in a lightweight and unsupervised manner is new**. Specifically, we measure the discriminativeness of old gallery features via feeding them into the trained and fixed new classifier without the requirements of any gallery classes. Such an operation showcases the great potential for further work that incorporates old features with the new model even though they have non-overlapping classes.
>
> More importantly, since the uncertainty-based backfilling strategy attempts to refresh the poor features (which show a higher risk for negative flips) in priority, it can **indirectly alleviate the model regression problem**, contributing to our main task of hot-refresh model upgrades, as shown in Figure 6 & 11 (Page 9, 14).

---

> > ### Author Response · Authors · 2021-11-19
> > **Responses to Reviewer Fu6z (2/3)**
> >
> > **Q2:** The other main contribution is in the Regression-Free Compatibility Regularization (RFCT) in Eq. 6, ..., as this simple addition to the loss could have a lot more variety resulting in much more in-depth analysis of the impact of this term - e.g.
> >
> > **(a)** changing the ratio between the two terms in the loss?
> >
> > **(b)** Or perhaps different temperatures in each of them?
> >
> > **(c)** Or even some dynamic tuning of these hyperparameters according to the uncertainty in the backfilling process, which ties the two novelties together. I think there is so much more that can be done to strengthen this novelty, but unfortunately is not presented in this work.
> >
> >
> >
> > **A2:**
> >
> > We really appreciate the valuable comments for improving our method. First, we would like to emphasize that two techniques of uncertainty-based backfilling and regression-alleviating loss function are not the only contributions of our work. More importantly, we
> > (i) for the first introduce a practical hot-refresh model upgrading mechanism;
> > (ii) observe the problem of model regression in such a new upgrading mechanism and analyze the essential reasons for it, i.e., negative flips;
> > (iii) introduce a simple yet effective method as the first attempt to tackle such a new problem. Although taking new-to-new negative pairs into the loss seems a simple addition, it is pretty essential to alleviating the negative flips, without which the model regression problem cannot be solved to the degree as it was in our method (see "vanilla" v.s. "ours" in Figures 3/4/5 of the main paper and Figures 8/9/10 in Appendix).
> >
> >
> >
> > **(a)** Following your advice, we investigate the trade-off between the new-to-old negatives and new-to-new negatives in Appendix A.4 (Page 17). Since it does not make sense to directly impose weighted factors on the sample distances (i.e., $\langle\cdot,\cdot\rangle$) in our original loss function (Eq. (6)), we first split our original unified contrastive loss into a combination of two losses  (${\mathcal{L}}_\text{ra-comp-split}(x) $, Eq.(13) in the main paper).
> > The results can be found in Figure 14 of Appendix. When setting $\eta=1.0$, similar results can be observed as the original unified contrastive loss. When changing $\eta$ from 0.2 to 5.0, we find that **a larger $\eta$ may lead to better alleviation of model regression but lower new-to-old compatible performance (0%)**. We find **$\eta=1.0$ a good trade-off** between compatibility and model regression, showing that the design of our unified version (Eq. (6)) of contrastive loss is reasonable.
> >
> >
> >
> > **(b)** To further analyze the effects of different temperatures in two types of negative pairs, we conduct experiments as shown in Table 7 of Appendix. Since similar results can be observed when changing the temperature, so we use 0.05 for both for brevity.
> >
> >
> >
> > **(c)** Thanks for the valuable suggestions. We agree that dynamic tuning of these hyper-parameters using the uncertainty metrics may further improve our method, but it is not trivial to be added in the rebuttal phase. We will try it in future works. Plus, the two novelties of regression-alleviating regularization and uncertainty-based backfilling are actually not independent in our work. They together contribute to alleviating the model regression problem. The difference is that, the former **explicitly** regularizes the models to reduce negative flips at training time, while the latter **implicitly** reduce negative flips by refreshing the poor features (which show a higher risk for negative flips) in priority at test time. They can be considered as **an entire solution** (from training to testing) for hot-refresh model upgrades.

---

> > > ### Author Response · Authors · 2021-11-19
> > > **Responses to Reviewer Fu6z (3/3)**
> > >
> > > **Q3:**  I think in this case, claiming the main contribution of this work as “regression-free” is slightly misleading and I would suggest toning down the name to something less strong, e.g. “regression-alleviating”.
> > >
> > >
> > >
> > > **A3:** Thanks for the valuable advice. We have changed the designation from "regression-free" to "regression-alleviating" in our revised manuscript.
> > >
> > >
> > >
> > >
> > >
> > > **Q4:** As the authors also took inspiration from active learning, I am wondering how does “hot-fresh” model upgrades compare against other techniques such as continual learning? For example, Continual Learning for Image-Based Camera Localization (Wang et. al 2021) also tackles the problem of learning gradually on unseen data, but on the problem of visual localization (which is still quite related to image retrieval). As I am not an expert on active learning / model upgrades, it would be great if the authors could clarify this.
> > >
> > >
> > >
> > > **A4:**
> > > **First**, we would like to discuss the **tasks** of hot-refresh model upgrades and continual learning, which differ in both applications and purposes.
> > >
> > > (i) Application: Hot-refresh model upgrade requires feature compatibility for **cross-model retrieval**, i.e., the query is embedded by the new model while the gallery is a mix of both new and old features. Continual learning requires the new model to remember the "capability" of the old model task during training, however, the new model is **individually deployed** without the "interaction" with the old model.
> > >
> > > (ii) Purpose: Hot-refresh model upgrade **requires** the new model to **outperform** the old model and the new features to be interoperable with old features. Continual learning only requires the new model to inherit the old task knowledge, **does not expect** the new model to **outperform** the old model on the old task.
> > >
> > > **Then**, regarding the **algorithms** that aim to tackle the above two tasks, we argue that *since continual learning does not need to conduct cross-model deployment, there's no consideration of model regression in existing continual learning methods*. To verify that existing continual learning algorithms cannot tackle the model regression problem of hot-refresh model upgrades, we compared to related work, namely Learning without Forget [1]. As shown in Figure 18 of Appendix, significant model regression and inferior feature compatibility are observed.
> > >
> > >
> > > Ref:
> > >
> > > [1] Li Z, Hoiem D. Learning without forgetting[J]. IEEE transactions on pattern analysis and machine intelligence, 2017.

---

> > > > ### Author Response · Authors · 2021-11-27
> > > > **Looking forward to your reply and the discussions!**
> > > >
> > > > Dear Reviewer Fu6z,
> > > >
> > > > Thank you again for your detailed comments and valuable suggestions! Regarding your main concerns in the initial comments,
> > > >
> > > > (1) *“the lack of exploring more on its main contribution, which is the RFCT loss, and potentially giving more valuable information to the community on the synergies between old and new features in the model upgrade process”:*
> > > >
> > > > Actually, our contributions are not limited to the RFCT loss.
> > > >
> > > > We **for the first time** introduce the task of hot-refresh model upgrades, which itself is a novel and valuable application of leveraging the synergies between old and new features. **The contribution of this newly proposed task** has been well acknowledged by the other reviewers.
> > > >
> > > > The RFCT loss acts as the first attempt to solve the problem of model regression in this newly proposed task, which is simple but effective. Note that **the observation of the model regression problem is also another important contribution** in our work, also acknowledged by the other reviewers.
> > > >
> > > > We agree that there exists more potential with the RFCT loss and appreciate your suggestions on improving it. Following your advice, we conducted experiments on analyzing the trade-off between the ratios of two negative terms and the effects of temperature hyper-parameters. **The results and discussions have been supplemented in the revised manuscript.**
> > > >
> > > > (2) *“the slight overclaim in the "Regression-Free" naming requires some extra attention”:*
> > > >
> > > > We really appreciate this constructive suggestion and have modified the designation of our method throughout the manuscript, i.e., "regression-free" to "regression-alleviating".
> > > >
> > > > We are wondering if our responses and additional experiments have well addressed your concerns. Looking forward to your reply and the discussions!
> > > >
> > > >
> > > > Best Regards,
> > > >
> > > > ICLR 2022 Conference Paper231 Authors

---

> > > > > ### Comment · Reviewer_Fu6z · 2021-11-28
> > > > > **Boosting initial score to "6: marginally above the acceptance threshold"**
> > > > >
> > > > > Dear Authors,
> > > > >
> > > > > Thank you very much for your detailed responses to my initial review. After careful examination of other reviewers' comments and your revised draft, I am convinced that you have sufficiently addressed my concerns. Specifically,
> > > > > - I am satisfied with the change from 'Regression-Free' to 'Regression-Alleviating'.
> > > > > - Given the better context (clarification of the difference between model upgrades and continual learning *v.s.* model upgrades and from other reviewer's acknowledgment of the contributions), I correct my initial stance that the contribution is lacking, as the proposition of hot-refresh upgrades and discovery of model regression are sufficiently novel and would benefit the community significantly in my opinion.
> > > > > - The extra study of the parameters in the RFCT loss also strengthens this contribution and is a good start, although I still think there is more potential to expand upon it (as does reviewer QLFh in the discussion I believe).
> > > > >
> > > > > Given the above, I am improving the score to "6: marginally above the acceptance threshold" as promised in my initial review and hope this paper would be presented at ICLR2022 for the image retrieval community. Many thanks for the effort!

---

### Official Review · Reviewer_6wRf · 2021-11-02

**Correctness:** 3
**Technical Novelty And Significance:** 3
**Empirical Novelty And Significance:** 3
**Recommendation:** 6
**Confidence:** 4

**Details Of Ethics Concerns:**

The reviewer has no concerns.

**Main Review:**

STRENGTHS:
1) The problem is very interesting and is not limited to social network applications. Compatible representations basically decouple learning from matching so that up-to-date features can always be used to match to the gallery-set. This may have several advantages especially in all those applications in which back-propagation is considered too slow to provide up to date features.
2) Hot-refresh is a novel problem.

WEAKNESSES:
1) Contrastive loss seems to be a distillation loss because $\phi_{\rm old}$ is a fixed network. This should be better discussed.
2) The paper seems to be divided into two independent parts. The first one is related to the combination of compatible learning with the reduction of negative flips, the second one is about refreshing the gallery. Their dependency seems to be only related because of the final application. This should be discussed.
3) Contrastive learning usage is not motivated. For example, feature learning through a surrogate classification task is typically performing better than contrastive learning. This is also shown in the landmark dataset paper (Weyand et al., 2020), in which the cosface method is used. Moreover the contrastive effect of the loss seems to be limited being $\phi_{\rm old}$ a fixed network. This part is less developed with respect to the rest of the paper. Compatibility is also less evaluated.
4) Sequential multi-model compatibility in upgrading to novel fresh data is not evaluated. This would have given the paper a lot more value. Even if sequential multimodal compatibility did not perform well, it would have been a great baseline to compare with hot-refresh model update. For example a five step sequential-upgrades with 20%, 40%, 60%, 80% and 100% of the data should be evaluated.
5) A practical disadvantage of hot-refreshing through resampling is that the original image gallery has to be stored. Compatible learning typically does not require storing the original images. In many applications original images cannot be stored for privacy issues.
6) The resampling strategy based on discriminative uncertainty seems to require multiple instances of the same classes in the gallery. This could be a strong assumption since the gallery is not a statistically meaningful set like for example the test-set in classification tasks.

**Summary Of The Paper:**

The paper presents a method to learn compatible features with low negative flips rate. The paper also introduces a resampling strategy to gradually replace the gallery to ameliorate the performance. Resampling is based on feature uncertainty. The less discriminative a feature is, the sooner it is replaced.

**Summary Of The Review:**

The paper introduces a very nice problem with some technical details to be clarified. The reviewer's opinion is that paper is not "uniformly" developed, many technical parts would have required more attention.

---

> ### Author Response · Authors · 2021-11-19
> **Responses to Reviewer 6wRf (1/3)**
>
> Thank you for the valuable and positive comments! We really appreciate the comments for improving the clarity of statements and experimental verifications. The manuscript is revised accordingly, and the main concerns are listed below.
>
> **Q1:** Contrastive loss seems to be a distillation loss because $\phi_\text{old}$ is a fixed network.
>
>
>
> **A1:**  Although we both use a fixed old model, we have entirely different **purposes**. Specifically, distillation aims at training a new model that inherits the knowledge of the teacher (old) model in order to improve the overall accuracy of the new model, and the new model will be deployed individually during inference. In contrast, our task requires cross-model retrieval via learning compatible features at the same time reducing negative flips. Well noted that learning new features that are interoperable with old features (as in our task) is **not equal to** inheriting the knowledge of old features (as in knowledge distillation).
>
> Moreover, we have different **experimental setups**. In knowledge distillation, the old model is generally a **stronger** network with more powerful representation discrimination. In our model upgrading task, the old model is generally a **poorer** network with inferior representation discrimination.
>
> To verify that existing distillation-based algorithms are inapplicable for hot-refresh model upgrades of retrieval tasks, we compared to related work, namely Focal Distillation [1], which was specially designed for alleviating the model regression of classification tasks via knowledge distillation. As shown in Figure 16 and 17 of Appendix, directly employing Focal Distillation in our task achieves unsatisfactory retrieval performances and is hard to achieve the goal of model upgrades. It is actually due to the fact that **it inherits the knowledge of the inferior old model too much**, which limits its capability of learning discriminative new features. We have added the detailed discussions in Appendix A.5, marked in blue.
>
>
>
> Ref: [1] Sijie Yan, Yuanjun Xiong, Kaustav Kundu, Shuo Yang, Siqi Deng, Meng Wang, Wei Xia, and Stefano Soatto. Positive-congruent training: Towards regression-free model updates. CVPR 2021.
>
>
>
> **Q2:**  The paper seems to be divided into two independent parts. The first one is related to the combination of compatible learning with the reduction of negative flips, the second one is about refreshing the gallery. Their dependency seems to be only related because of the final application. This should be discussed.
>
>
>
> **A2:** We tackle the problem of model regression in hot-refresh model upgrades. The introduced two technical parts, i.e., regression-free compatible training and uncertainty-based backfilling strategy, **together contribute to alleviating this problem**. The difference is that, the former **explicitly** regularizes the models to reduce negative flips at training time, while the latter **implicitly** reduce negative flips by refreshing the poor features (which show a higher risk for negative flips) in priority at test time. Plus, they are not technically independent, since the proposed backfilling strategy requires the new classifier for producing the class probabilities of the old gallery features. They can be considered as **an entire solution** (from training to testing) for hot-refresh model upgrades. To make an analogy, their relations are similar to the model training and re-ranking post-processing for conventional retrieval tasks. We added the discussions in the method part in blue.

---

> > ### Author Response · Authors · 2021-11-19
> > **Responses to Reviewer 6wRf (2/3)**
> >
> > **Q3:**
> >
> > **(a)** Contrastive learning usage is not motivated. For example, feature learning through a surrogate classification task is typically performing better than contrastive learning. This is also shown in the landmark dataset paper (Weyand et al., 2020), in which the cosface method is used.
> >
> > **(b)** Moreover the contrastive effect of the loss seems to be limited being a fixed network. This part is less developed with respect to the rest of the paper.
> >
> > **(c)** Compatibility is also less evaluated.
> >
> >
> >
> >
> > **A3:**
> >
> > **(a)** We use the form of contrastive learning in order to **explicitly** enforce the new-to-old positive pairs to be more similar than both the new-to-old negative pairs and the new-to-new negative pairs. Contrastive loss is also used in related work [2]. Actually the form of compatible constraints is not the core of this work, instead, the core is to take new-to-new negative pairs into consideration for reducing negative flips.
> >
> > We agree that using classification as a pretext task may benefit feature learning, e.g., BCT [3] obtains slightly better compatibility than ours in Figure 16 (a)&(b) (see 0% for new-to-old compatibility), but fails to alleviate the problem of model regression. BCT feeds the new features into the old classifier to force new features to be closer to the old class center. It is hard to explicitly impose the regularization of new-to-new negative pairs into the form of BCT (classification) loss. So we try to use our regression-alleviating compatible loss on top of BCT loss, and achieve state-of-the-art compatible performance at the same time alleviating model regression (see Figure 16 & 17 of Appendix).
> >
> > Thanks for the valuable advice. We have added explanations and comparisons in the manuscript, as shown in Appendix A.5, marked in blue.
> >
> >
> >
> > **(b)** Sorry for the confusion. The fixed old model will be used whatever kinds of supervision are chosen in our task. So the fixed network would not be a factor that hinders the effects of contrastive loss. It is a common setup to freeze the old model in compatible learning tasks [2,3], also it satisfies the real-world situation of model upgrades.
> >
> >
> >
> > **(c)** The evaluation of compatibility can be observed at 0% in Figures 3/4/5 of the main paper and Figures 8/9/10 of Appendix. Specifically, at 0% of model upgrades, the new features of queries need to be directly indexed by the old gallery feature, which is an ordinary benchmark for compatible learning. We can observe that our method can always achieve better compatible performance than the vanilla baseline, indicating that we do not sacrifice the compatibility for reducing negative flips. For more intuitive comparisons, we listed the above results in Table 2 of Appendix.
> >
> >
> >
> > | Model              | Expansion | Expansion      | Expansion | Open-data |         Open-data       | Open-data |       Open-class         |         Open-class       |      Open-class          |
> > | ------------------ | -------------- | -------------- | -------------- | -------------- | -------------- | -------------- | -------------- | -------------- | -------------- |
> > |                    | GLDv2          | ROxford        | RParis         | GLDv2          | ROxford        | RParis         |       GLDv2         |        ROxford        |      RParis          |
> > | R50 (old)          | 9.91           | 64.28          | 83.27          | 9.91           | 64.28          | 83.27          | 11.04          | 64.40          | 83.40          |
> > | R50 (vanilla new)  | 10.95          | 64.23          | **79.68** | 10.46          | 62.46          | 78.96          | 11.70          | 63.86          | 81.48          |
> > | R50 (ours new)     | **11.13** | **67.34** | 79.65          | **10.76** | **64.38** | **81.63** | **11.97** | **65.10** | **83.60** |
> > | R101 (vanilla new) | 11.37          | 65.87          | 81.52          | 10.61          | 63.89          | 79.31          | 11.65          | 63.86          | 81.52          |
> > | R101 (ours new)    | **11.40** | **66.54** | **83.75** | **11.05** | **64.33** | **81.64** | **11.90** | **65.10** | **83.61** |
> >
> >
> >
> > Ref: [2] Mateusz Budnik and Yannis Avrithis. Asymmetric metric learning for knowledge transfer. CVPR 2021.
> >
> > [3] Yantao Shen, Yuanjun Xiong, Wei Xia, and Stefano Soatto. Towards backward-compatible representation learning. CVPR 2020.

---

> > > ### Author Response · Authors · 2021-11-19
> > > **Responses to Reviewer 6wRf (3/3)**
> > >
> > > **Q4:** Sequential multi-model compatibility in upgrading to novel fresh data is not evaluated. This would have given the paper a lot more value. Even if sequential multimodal compatibility did not perform well, it would have been a great baseline to compare with hot-refresh model update. For example a five step sequential-upgrades with 20%, 40%, 60%, 80% and 100% of the data should be evaluated.
> > >
> > >
> > >
> > > **A**4: Thanks for the constructive suggestion!
> > >
> > > First, we would like to clarify that our hot-refresh model upgrades are **not in conflict** with sequential multi-model compatibility. To be specific, previous compatible works conduct the backfill-free model upgrades via increasing the retrieval performance from "old-to-old" to "new-to-old" retrieval (in the form of "query-to-gallery"), dubbed as "no-refresh" model upgrades in this paper. Our introduced "hot-refresh" model upgrades conducts "old-to-old" -> "new-to-old" -> "new-to-new", boosting the upper bound for the retrieval system via further refreshing the gallery in a hot-refresh manner. Sequential multi-model compatibility can also been applied with our hot-refresh model upgrading mechanism.
> > >
> > > We conducted the sequential upgrade experiments following your advice (see Figure 7 of  the main paper, Section5.4). Specifically, we maintain the same old model (trained with 30% data) as we used in the other parts of the paper for fair comparison, and then we train the other three new models by using 60% (1st generation), 80% (2nd generation) and 100% (3rd generation) data, respectively. Sequential model upgrades in our hot-refresh manner significantly outperform the no-refresh manner. It is actually due to the fact that the poor discriminativeness of the old gallery features highly limits the retrieval accuracy of the new model in no-refresh model upgrades.
> > >
> > >
> > >
> > >
> > >
> > > **Q5:** A practical disadvantage of hot-refreshing through resampling is that the original image gallery has to be stored. Compatible learning typically does not require storing the original images. In many applications original images cannot be stored for privacy issues.
> > >
> > >
> > >
> > > **A5:** Yes, we acknowledge that the requirement for the original images is a limitation for our proposed hot-refresh model upgrades. However, we cannot neglect the benefits of hot-refresh model upgrades in applications that have raw images. Recall that our hot-refresh model upgrade achieves significantly better performance than no-refresh model upgrade (see A4). We carefully discussed this limitation in Section 6 in blue. We considered forward-compatible learning as a potential solution for dealing with this limitation, which is expected to upgrade the gallery features with the input of only the old features rather than raw images. How to properly alleviate the problem of model regression when conducting hot-refresh model upgrades with forward-compatible learning is also an important task and has never been studied. Further investigation is called for.
> > >
> > >
> > >
> > >
> > >
> > > **Q6:** The resampling strategy based on discriminative uncertainty seems to require multiple instances of the same classes in the gallery. This could be a strong assumption since the gallery is not a statistically meaningful set like for example the test-set in classification tasks.
> > >
> > >
> > >
> > > **A6:** Sorry for the confusion. We do not need any labels for the gallery set and have no restrictions or requirements on the distributions of the gallery. We simply feed the massive gallery into the trained and fixed classifier of the new model for inference under the assumption that the old features lay on the similar latent space as the new features. Although the ground-truth classes of the gallery images may not exist in the new classifier, we consider that more discriminative old features should have sharper class probability vectors.

---

> > > > ### Author Response · Authors · 2021-11-27
> > > > **Looking forward to your reply and the discussions!**
> > > >
> > > > Dear Reviewer 6wRf,
> > > >
> > > > Thank you again for your valuable comments and the acknowledgment of our contributions!
> > > >
> > > > Regarding your main concerns in the initial comments, we
> > > > (1) discussed the differences from knowledge distillation,
> > > > (2) clarified the connections between two technical parts, i.e., regression-free compatible training and uncertainty-based backfilling strategy,
> > > > (3) clarified the motivations of chosen loss formulations,
> > > > (4) made the evaluation of feature compatibility more clear,
> > > > (5) supplemented the experiments of sequential model upgrades,
> > > > (6) carefully discussed the limitations of hot-refresh model upgrades due to privacy issues,
> > > > (7) clarified the design of our backfilling strategy.
> > > >
> > > > We are wondering if our responses and additional experiments have well addressed your concerns. Looking forward to your reply and the discussions!
> > > >
> > > > Best Regards,
> > > >
> > > > ICLR 2022 Conference Paper231 Authors

---

### Official Review · Reviewer_JYkL · 2021-11-03

**Correctness:** 4
**Technical Novelty And Significance:** 3
**Empirical Novelty And Significance:** 2
**Recommendation:** 6
**Confidence:** 5

**Main Review:**

_Strengths_

- The presentation of the paper is very good. The paper is well written and the method has been properly motivated. The quality of the figures is also very good and they help a lot to understand the problem, the method, and the experiments.
- It addresses an interesting problem that has real-world application in industrial image retrieval applications

---
_Weaknesses_

- Although it addresses a novel problem in image retrieval, I find the novelty of the method itself somehow limited. It's very similar to other methods that sample pairs of features coming from the new and the old model, being the main contribution in this case that new-to-new negative pairs are also considered.
- I have a couple of comments related to the fact that the authors decided not to use new-to-new positive pairs in the "regression-free compatible loss" (Lcomp and Lrf-comp). First of all, what I understand from the main paper is that they decided not to do it because "it is hard to make sure intra-class instances exist in the same mini-batch without a pre-sampler", however, adding this pre-sampler is actually easy and trivial, hence it is doable and not a reason why not to consider it in the first place. Secondly, the authors actually include an experiment in the appendix where they introduce a positive sampler and they show that similar results can be obtained by either using it or not. My suggestion here it is two fold: first I would suggest the authors to reference this experiment in the main paper to strength the decision of not including positive pairs. Second, I wondered if the reason of not seeing an improvement when sampling positives pairs in the batch it has to do with the fact that they're seen as negatives (?) in the second term of the denominator in Lrf-comp (k \in B/x). Could other losses be more suitable for this scenario, eg. a triplet loss where several triplets combining positives and negatives from "new" and "old" are considered at the same time?
- I found Figure 6 a bit difficult to "read" and to draw conclusions from it. Maybe showing an average per dataset or protocol would be better? Not sure about this though.

**Summary Of The Paper:**

The paper addresses the task of hot-refresh model upgrade (ie. the task of deploying a new model into production while the gallery features from an older are still being re-indexed) in retrieval systems and studies the problem of model regression that happens when the similarity of new-feature-to-old-feature positive pairs is lower than new-feature-to-new-feature negative pairs. To solve this the authors propose a compatible training method that takes these flips into account, encouraging the new-to-old pair positives to be more similar than both new-to-old and new-to-new negative pairs (this is nicely illustrated in Figure 2). The paper also introduces an uncertainty-based backfilling (ie. the process of re-indexing the old gallery features with the new model) strategy that tries to follow a poor-first policy using the probability distribution of the training classes yielded by the new classifier.

**Summary Of The Review:**

Even though I consider the technical contribution a bit limited I positively acknowledge the fact that they're the fist to study the problem of hot-refresh model upgrade in image retrieval in particular and the problem of model regression.

---

> ### Author Response · Authors · 2021-11-19
> **Responses to Reviewer JYkL (1/3)**
>
> Thank you for the constructive and positive comments! We really appreciate the comments for improving the clarity of statements and experimental verifications. The manuscript is revised accordingly, and the responses to your main concerns are listed below.
>
> **Q1**: Although it addresses a novel problem in image retrieval, I find the novelty of the method itself somehow limited. It's very similar to other methods that sample pairs of features coming from the new and the old model, being the main contribution in this case that new-to-new negative pairs are also considered.
>
>
>
> **A1**: Thank you first for acknowledging the novelty of the introduced problem.
>
> We would like to emphasize that existing compatible learning methods that exploit new and old paired features did not consider the task of hot-refresh model upgrades thus overlooked the problem of negative flips. Therefore, directly employing these methods cannot alleviate the model regression problem (for example, see "BCT" in Figures 16/17 of Appendix).
>
> Although taking new-to-new negative pairs into consideration seems a small change to existing compatible training methods, it is pretty **essential to alleviating the negative flips**, without which the model regression problem cannot be solved to the degree as it was in our method (see "vanilla" v.s. "ours" in Figures 3/4/5 of the main paper and Figures 8/9/10 of Appendix).
>
> Recall that studying advanced compatibility constraints is not the core of our work, instead, our main contribution is to:
>
> (i) first introduce a practical hot-refresh model upgrading mechanism, which is made possible by compatible feature learning;
>
> (ii) observe the problem of model regression in such an upgrading mechanism and analyze the reasons for it, i.e., negative flips;
>
> (iii) introduce a simple yet effective method as the first attempt to tackle such a new problem, which showcases the great potential for further work that incorporates the consideration of new-to-new negative pairs with more advanced compatible regularizations.

---

> > ### Author Response · Authors · 2021-11-19
> > **Responses to Reviewer JYkL (2/3)**
> >
> > **Q2:** I have a couple of comments related to the fact that the authors decided not to use new-to-new (**we guess it is a typo, and should be "new-to-old"**) positive pairs in the "regression-free compatible loss" (Lcomp and Lrf-comp). First of all, what I understand from the main paper is that ... Secondly, the authors actually include an experiment in the appendix ... My suggestion here it is two fold:
> >
> > **(a)** First I would suggest the authors to reference this experiment in the main paper to strength the decision of not including positive pairs.
> >
> > **(b)** Second, I wondered if the reason of not seeing an improvement when sampling positives pairs in the batch it has to do with the fact that they're seen as negatives (?) in the second term of the denominator in Lrf-comp (k \in B/x).
> >
> > **(c)** Could other losses be more suitable for this scenario, eg. a triplet loss where several triplets combining positives and negatives from "new" and "old" are considered at the same time?
> >
> >
> >
> > **A2:**
> >
> > **(a)** Thanks for the constructive suggestion. We have revised the manuscript and referenced this experiment in the main paper (marked as blue in Section 4.1).
> >
> > **(b)** Sorry for the confusion about the denominator of $\mathcal{L}_\text{rf-comp}$, we would like to clarify that we had filtered out the intra-class pairs of the anchor image in the denominator in all of our experiments, so the positives would not be treated as the negatives.  Our pseudo code in Page 10 also illustrates the details (realized by the "masks" in Algorithm 1). We have made the equations (Eq. (5)&(6)) more clear, i.e., changing ${\mathcal{B}}$\\ $x$ to ${\mathcal{B}}$\\ $p(x)$, where $p(x)$ denotes the intra-class samples for $x$ (including $x$ itself).
> >
> > The reason for not seeing an improvement with the positive sampler might be that, in the task of compatible training, the most significant objective is to train the new model to map images to same features as the old model, rather than constraining the distance between positive pairs. Such a phenomenon has also been observed by the related work [1]. As shown in Table 3 of [1], the method of "regression" (forces the old feature and new feature of the same image similar using cosine similarity) wins the asymmetric test (cross-model test), compared to other methods that regularize the positive pairs. Moreover, the training dataset, GLDv2, consists of 1,580,470 samples with 81,313 classes, i.e., \~19 samples per class, showing limited intra-class variations, which might also lead to similar results with or without such a positive sampler. We have added the detailed explanations in the Appendix A.4 (Table 6, Page 17).
> >
> > **(c)** We supplemented the experiments using the form of triplet loss in the Appendix A.4 (Figure 15, Page 18). We observe that our method can still effectively reduce negative flips on GLDv2-test and RParis datasets, but fail in ROxford. It is mainly due to the fact that the unified form of a contrastive loss can adaptively focus on the hard negatives among new-to-old and new-to-new pairs while the triplet loss cannot. We have also investigated the effects of the positive sampler with triplet loss, as demonstrated in Table 8 of Appendix. The same conclusion as the contrastive counterparts is drawn, i.e., using the sampler or not achieves similar results. So the reasons for not seeing an improvement should lay on the objectives of compatible training and the dataset characteristics (as we discussed in (**b**)) rather than the form of the regularizations.
> >
> >
> >
> > Ref: [1] Mateusz Budnik and Yannis Avrithis. Asymmetric metric learning for knowledge transfer. CVPR 2021.

---

> > > ### Author Response · Authors · 2021-11-19
> > > **Responses to Reviewer JYkL (3/3)**
> > >
> > > **Q3**:  I found Figure 6 a bit difficult to "read" and to draw conclusions from it. Maybe showing an average per dataset or protocol would be better?
> > >
> > >
> > >
> > > **A3**: Thanks for the valuable suggestion. We took different uncertainty-based backfilling strategies apart, as illustrated in Figure 6 of the main paper and Figures 11/12 of Appendix in the revised manuscript. We observed that,
> > >
> > > **(i)** Least Confidence and Margin of Confidence show more stable benefits compared to Entropy-based method, indicating that the probabilities of top-2 confident classes are more important than the overall classes;
> > >
> > > **(ii)** The uncertainty-based backfilling strategy can fasten the mAP improvements on top of either vanilla method or our method;
> > >
> > > **(iii)** Since the uncertainty-based backfilling strategy attempts to refresh the poor features (which show a higher risk for negative flips) in priority, it can **indirectly alleviate the model regression problem**.
> > >
> > > However, we still find some failure cases, as demonstrated in Figure 13 of Appendix. The uncertainty-based strategy may not work when the new and old features are not compatible enough, since we use the trained and fixed new classifier to produce class probabilities for the old gallery features under the assumption that they lay on the almost same latent space. To further investigate the phenomenon of Figure 13, we observe that the new-to-old performance (0%) is inferior to the old-to-old performance (brown star) in Figure 5(b) of the main paper, showing the unsatisfactory feature compatibility. We carefully discuss such a limitation for our introduced uncertainty-based backfilling strategy in Appendix A.3 (Page 14-15).

---

> > > > ### Author Response · Authors · 2021-11-27
> > > > **Looking forward to your reply and the discussions!**
> > > >
> > > > Dear Reviewer JYkL,
> > > >
> > > > Thank you again for the acknowledgment of our contributions and the positive comments!
> > > >
> > > > Regarding your main concerns in the initial comments, we
> > > > (1) clarified the technical contribution (the small change of the compatible learning objectives seems simple but is pretty essential to alleviating the model regression; this success attempt showcases the great potential for further work that incorporates the consideration of new-to-new negative pairs with more advanced compatible regularizations),
> > > > (2) added more analysis on the positive sampler, clarified the reasons about why it does not work, and supplemented the experiments of triplet loss,
> > > > (3) made the figures and conclusions of the uncertainty-based backfilling strategy more clear.
> > > >
> > > > We are wondering if our responses and additional experiments have well addressed your concerns. Looking forward to your reply and the discussions!
> > > >
> > > > Best Regards,
> > > >
> > > > ICLR 2022 Conference Paper231 Authors

---

### Decision · Program_Chairs · 2022-01-20

**Decision:**

Accept (Poster)

**Comment:**

This paper is proposed to deeply investigate the hot-refresh model upgrades of image retrieval systems. The hot-refresh model is very useful since the model can be quickly updated after the gallery is backfilled. To address the model regression with negative flips, this paper introduces a Regression-Alleviating Compatible Training (RACT) method by reducing negative flips. The proposed method has been verified on the large-scale image retrieval benchmark such as Google Landmark. The key contribution of this paper is the new setting targeting an important application in real-world image retrieval systems. However, some of the technical details are not fully explained. Despite these minor concerns, the AC will rate this paper as a poster acceptance based on the overall contributions.